# MATCHA: MITIGATING GRAPH STRUCTURE SHIFTS WITH TEST-TIME ADAPTATION

**Wenxuan Bao[1], Zhichen Zeng[1], Zhining Liu[1], Hanghang Tong[1], Jingrui He[1]**
[1]University of Illinois Urbana-Champaign
{wbao4,zhichenz,liu326,htong,jingrui}@illinois.edu

## ABSTRACT

Powerful as they are, graph neural networks (GNNs) are known to be vulnerable to distribution shifts. Recently, test-time adaptation (TTA) has attracted attention due to its ability to adapt a pre-trained model to a target domain, without re-accessing the source domain. However, existing TTA algorithms are primarily designed for attribute shifts in vision tasks, where samples are independent. These methods perform poorly on graph data that experience structure shifts, where node connectivity differs between source and target graphs. We attribute this performance gap to the distinct impact of node attribute shifts versus graph structure shifts: the latter significantly degrades the quality of node representations and blurs the boundaries between different node categories. To address structure shifts in graphs, we propose `Matcha`, an innovative framework designed for effective and efficient adaptation to structure shifts by adjusting the hop-aggregation parameters in GNNs. To enhance the representation quality, we design a prediction-informed clustering loss to encourage the formation of distinct clusters for different node categories. Additionally, `Matcha` seamlessly integrates with existing TTA algorithms, allowing it to handle attribute shifts effectively while improving overall performance under combined structure and attribute shifts. We validate the effectiveness of `Matcha` on both synthetic and real-world datasets, demonstrating its robustness across various combinations of structure and attribute shifts. Our code is available at `https://github.com/baowenxuan/Matcha`.

## 1 INTRODUCTION

Graph neural networks (GNNs) have shown great success in various graph applications such as social networks (Rozemberczki et al., 2021), scientific literature networks (Hu et al., 2020), and financial fraud detection (Pareja et al., 2020). Their success heavily relies on the assumption that training and testing graphs are identically distributed (Li et al., 2022a). However, real-world graphs usually involve distribution shifts in both node attributes and graph structures (Liu et al., 2023; Wu et al., 2023a;b). For example, given two social networks (e.g., LinkedIn for professional networking and Instagram for casual content sharing), the user profiles are likely to vary due to the different functionalities of two graphs, resulting in *attribute shifts*. Besides, as LinkedIn users tend to connect with professional colleges, while users on Instagram often connect with family and friends, the connectivity patterns vary across different networks, introducing *structure shifts*. The co-existence of these complex shifts significantly undermines GNN model performance (Li et al., 2022a).

Various approaches have been proposed to tackle distribution shifts between the source and target domains, e.g., domain adaptation (Wang & Deng, 2018) and domain generalization (Wang et al., 2023b). But most of these approaches require access to either target labels (Wu et al., 2023a;b) or the source domain during adaptation (Liu et al., 2023; Xiao et al., 2023), which is often impractical in real-world applications. For example, when a model is deployed for fraud detection, the original transaction data used for training may no longer be accessible. Test-time adaptation (TTA) has emerged as a promising solution, allowing models to adapt to an unlabeled target domain without re-accessing the source domain (Liang et al., 2023). These algorithms demonstrate robustness against various image corruptions and style shifts in vision tasks (Wang et al., 2021; Iwasawa & Matsuo, 2021; Zhang et al., 2023). However, applying TTA to graph data presents significant challenges, especially under structure shifts. As shown in Figure 1, both attribute and structure shifts (e.g.,

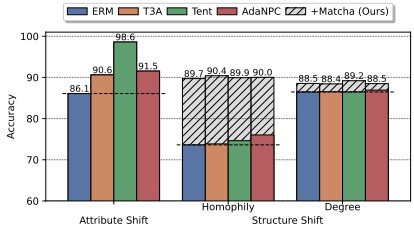 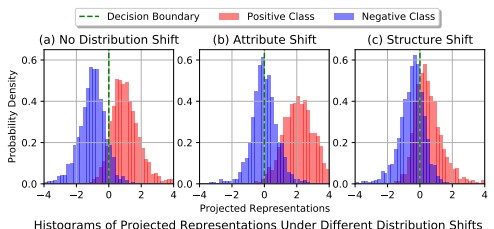

Figure 1: Generic TTA algorithms (T3A, Tent, AdaNPC) are significantly less effective under structure shifts (right) than attribute shifts (left). On the contrary, our proposed `Matcha` could significantly improve the performance of generic TTA (gray shaded area). The dataset used is CSBM.

Figure 2: Attribute shifts and structure shifts have different impact patterns. Compared to attribute shifts (b), structure shifts (c) mix the distributions of node representations from different classes, which cannot be alleviated by adapting the decision boundary. This explains the limitations of existing generic TTA algorithms. The dataset used is CSBM.

homophily and degree shifts) lead to performance drops on target graphs, but current TTA methods provide only marginal accuracy improvements under structure shifts compared to attribute shifts.

In this paper, we seek to understand why generic TTA algorithms perform poorly under structure shifts. Through both theoretical analysis and empirical evaluation, we reveal that while both attribute and structure shifts affect model accuracy, they impact GNNs in different ways. Attribute shifts mainly affect the decision boundary and can often be addressed by adapting the downstream classifier. In contrast, structure shifts degrade the upstream featurizer, causing node representations to mix and become less distinguishable, which significantly hampers performance. Figure 2 illustrates this distinction. Since most generic TTA algorithms rely on high-quality representations (Iwasawa & Matsuo, 2021; Wang et al., 2021; Zhang et al., 2023), they struggle to improve GNN performance under structure shifts.

To address these limitations, we propose that the key to mitigating structure shifts lies in restoring the quality of node representations, making the representations of different classes distinct again. Guided by theoretical insights, we propose adjusting the hop-aggregation parameters which control how GNNs integrate node features with neighbor information across different hops. Many GNN designs include such hop-aggregation parameters, e.g., GPRGNN (Chien et al., 2021), APPNP (Klicpera et al., 2019), JKNet (Xu et al., 2018), and GCNII (Chen et al., 2020). Building on this, we introduce `Matcha`, a framework to Mitigate grAph sTruCture sHifts with test-time Adaptation. It restores representation quality by adapting hop-aggregation parameters via minimizing prediction-informed clustering (PIC) loss, promoting discriminative node representations without falling into trivial solutions as with traditional entropy loss. Additionally, our framework can be seamlessly integrated with existing TTA algorithms to harness their capability to handle attribute shifts. We empirically evaluate `Matcha` with a wide range of datasets and TTA algorithms. Extensive experiments on both synthetic and real-world datasets show that `Matcha` can handle a variety of structure shifts, including homophily shifts and degree shifts. Moreover, it is compatible to a wide range of TTA algorithms and is able to enhance their performance under various combinations of attribute shifts and structure shifts. We summarize our contributions as follows:

- **Theoretical analysis** reveals the distinct impact patterns of attribute and structure shifts on GNNs, which limits the effectiveness of generic TTA methods in graphs. Compared to attribute shifts, structure shifts more significantly impair the node representation quality.
- **A novel framework** `Matcha` is proposed to restore the quality of node representations and boost existing TTA algorithms by adjusting the hop-aggregation parameters.
- **Empirical evaluation** on both synthetic and real-world scenarios demonstrates the effectiveness of `Matcha` under various distribution shifts. When applied alone, `Matcha` enhances the source model performance by up to 31.95%. When integrated with existing TTA methods, `Matcha` further boosts their performance by up to 40.61%.

## 2 RELATED WORKS

**Test-time adaptation** (TTA) aims to adapt a pre-trained model from the source domain to an unlabeled target domain without re-accessing the source domain during adaptation (Liang et al., 2023).

For i.i.d. data like images, several recent works propose to perform image TTA by entropy minimization (Wang et al., 2021; Zhang et al., 2022), pseudo-labeling (Iwasawa & Matsuo, 2021; Zhang et al., 2023), consistency regularization (Boudiaf et al., 2022), etc. However, graph TTA is more challenging due to the co-existence of attribute shifts and structure shifts. To address this issue, GTrans (Jin et al., 2023) proposes to refine the target graph at test time by minimizing a surrogate loss. SOGA (Mao et al., 2024) maximizes the mutual information between model inputs and outputs, and encourages consistency between neighboring or structurally similar nodes, but it is only applicable to homophilic graphs. Focusing on degree shift, GraphPatcher (Ju et al., 2023) learns to generate virtual nodes to improve the prediction on low-degree nodes. In addition, GAPGC (Chen et al., 2022) and GT3 (Wang et al., 2022) follow a self-supervised learning (SSL) scheme to fine-tune the pre-trained model for graph classification.

**Graph domain adaptation** (GDA) aims to transfer knowledge from a labeled source graph to an unlabeled target graph with access to *both* graphs. Most of the GDA algorithms focus on learning invariant representations over the source and target graphs by adversarial learning (Zhang et al., 2019; Wu et al., 2020; Xiao et al., 2023) or minimizing the distance between source and target graphs (Zhu et al., 2021b; Wu et al., 2023b). More recent works (Liu et al., 2023; 2024a) address the co-existence of structure and node attribute shifts by reweighing the edge weights of the source graphs. However, GDA methods require simultaneous access to both the source and target graphs, and thus cannot be extended to TTA scenarios.

We also discuss related works in (1) graph out-of-distribution generalization and (2) homophily-adaptive GNN models in Appendix E.1.

## 3 ANALYSIS

In this section, we explore how different types of distribution shifts affect GNN performance. We first introduce the concepts of attribute shifts and structure shifts in Subsection 3.1. Subsequently, in Subsection 3.2, we analyze how attribute shifts and structure shifts affect the GNN performance in different ways, which explain the limitation of generic TTA methods. Finally, in Subsection 3.3, we propose that adapting the hop-aggregation parameters can effectively handle structure shifts.

### 3.1 PRELIMINARIES

In this paper, we focus on graph test-time adaptation (GTTA) for node classification. A labeled source graph is denoted as $\mathcal{S} = (\boldsymbol{X}_S, \boldsymbol{A}_S)$ with node attribute matrix $\boldsymbol{X}_S \in \mathbb{R}^{N \times D}$ and adjacency matrix $\boldsymbol{A}_S \in \{0, 1\}^{N \times N}$. The corresponding node label matrix is denoted as $\boldsymbol{Y}_S \in \{0, 1\}^{N \times C}$. For a node $v_i$, we denote its neighbors as $\mathbb{N}(v_i)$ and node degree as $d_i$. A GNN model $g_S \circ f_S(\cdot)$ is pre-trained on the source graph, where $f_S$ is the featurizer extracting node-level representations, and $g_S$ is the classifier, which is usually a linear layer. The goal of GTTA is to adapt the pre-trained GNN model to enhance node classification accuracy on an unlabeled target graph $\mathcal{T} = (\boldsymbol{X}_T, \boldsymbol{A}_T)$ with a different distribution, while the source graph $\mathcal{S}$ are not accessible during adaptation.[1]

Compared with TTA on regular data like images, GTTA is more challenging due to the co-existence of *attribute shifts* and *structure shifts* (Li et al., 2022a; Wu et al., 2023b), which are formally defined as follows (Liu et al., 2023).

**Attribute shift.** We assume that the node attributes $\boldsymbol{x}_i$ for each node $v_i$ (given its label $\boldsymbol{y}_i$) are i.i.d. sampled from a class-conditioned distribution $\mathbb{P}_{\boldsymbol{x}|\boldsymbol{y}}$. The attribute shift is defined as $\mathbb{P}^{\mathcal{S}}_{\boldsymbol{x}|\boldsymbol{y}} \neq \mathbb{P}^{\mathcal{T}}_{\boldsymbol{x}|\boldsymbol{y}}$.

**Structure shift.** We consider the joint distribution of adjacency matrix and labels $\mathbb{P}_{\boldsymbol{A} \times \boldsymbol{Y}}$. The structure shift is defined as $\mathbb{P}^{\mathcal{S}}_{\boldsymbol{A} \times \boldsymbol{Y}} \neq \mathbb{P}^{\mathcal{T}}_{\boldsymbol{A} \times \boldsymbol{Y}}$. Specifically, we focus on two types of structure shifts: *degree shift* and *homophily shift*.

**Degree shift.** Degree shift refers to the difference in degree distribution, particularly the average degree, between the source graph and the target graph. For instance, in the context of a user co-purchase graph, in more mature business regions, the degree of each user node may be rela-

---

[1]This setting is also referred to as *source-free unsupervised graph domain adaptation* (Mao et al., 2024). Here, we primarily follow the terminology used by Jin et al. (2023). It is important to note that, unlike the online setting often adopted in image TTA, graph TTA allows simultaneous access to the entire unlabeled target graph $\mathcal{T}$ (Liang et al., 2023).

tively higher due to multiple purchases on the platform. However, when the company expands its operations to a new country where users are relatively new, the degree may be comparatively lower.

**Homophily shift.** Homophily refers to the phenomenon that a node tends to connect with nodes with the same labels. Formally, the *node homophily* of a graph $\mathcal{G}$ is defined as (Pei et al., 2020):

$$h(\mathcal{G}) = \frac{1}{N} \sum_i h_i, \quad \text{where } h_i = \frac{|\{v_j \in \mathbb{N}(v_i) : y_j = y_i\}|}{d_i}, \tag{1}$$

where $|\cdot|$ denotes the cardinality of a set. Homophily shift refers to the phenomenon that the source and target graphs have different levels of homophily. For example, with node labels as occupation, business social networks (e.g., LinkedIn) are likely to be more homophilic than other social networks (e.g., Pinterest, Instagram).

Although structure shifts do not directly change the distribution of each single node's attribute, they change the distribution of each node's neighbors, and thus affects the distribution of node representations encoded by GNNs.

## 3.2 IMPACTS OF DISTRIBUTION SHIFTS

As observed in Figure 1, both attribute shifts and structure shifts can impact the performance of GNNs. However, the same TTA algorithm demonstrates remarkably different behaviors when addressing these two types of shifts. We posit that this is due to the distinct ways in which attribute shifts and structure shifts affect GNN performance. We adopt the contextual stochastic block model (CSBM) and single-layer GCNs to elucidate these differences.

**CSBM** (Deshpande et al., 2018) is a random graph generator widely used in the analysis of GNNs (Ma et al., 2022; Mao et al., 2023; Yan et al., 2022). Specifically, we consider a CSBM with two classes $\mathbb{C}_+ = \{v_i : y_i = +1\}$ and $\mathbb{C}_- = \{v_i : y_i = -1\}$, each having $\frac{N}{2}$ nodes. The attributes for each node $v_i$ are independently sampled from a Gaussian distribution $\boldsymbol{x}_i \sim \mathcal{N}(\boldsymbol{\mu}_i, \boldsymbol{I})$, where $\boldsymbol{\mu}_i = \boldsymbol{\mu}_+$ for $v_i \in \mathbb{C}_+$ and $\boldsymbol{\mu}_i = \boldsymbol{\mu}_-$ for $v_i \in \mathbb{C}_-$. Each pair of nodes are connected with probability $p$ if they are from the same class, otherwise $q$. As a result, the average degree is $d = \frac{N(p+q)}{2}$ and node homophily is $h = \frac{p}{p+q}$. We denote the graph as $\text{CSBM}(\boldsymbol{\mu}_+, \boldsymbol{\mu}_-, d, h)$, where $\boldsymbol{\mu}_+, \boldsymbol{\mu}_-$ encode the node attributes and $d, h$ encode the graph structure.

**Single-layer GCN.** We consider a single-layer GCN, whose featurizer is denoted as $\boldsymbol{Z} = f(\boldsymbol{X}, \boldsymbol{A}) = \boldsymbol{X} + \gamma \cdot \boldsymbol{D}^{-1} \boldsymbol{A} \boldsymbol{X} = (\boldsymbol{I} + \gamma \cdot \boldsymbol{D}^{-1} \boldsymbol{A}) \boldsymbol{X}$, where $\boldsymbol{D}$ is the degree matrix. Equivalently, for each node $v_i$, its node representation is $\boldsymbol{z}_i = \boldsymbol{x}_i + \gamma \cdot \frac{1}{d_i} \sum_{v_j \in \mathbb{N}(v_i)} \boldsymbol{x}_j$. The parameter $\gamma$ controls the mixture between the node's own representation and its one-hop neighbors' average representation. We consider $\gamma$ as a fixed parameter for now, and adapt it later in Subsection 3.3. We consider a linear classifier as $g(\boldsymbol{Z}) = \boldsymbol{Z}\boldsymbol{w} + \boldsymbol{1}b$, which predicts a node $v_i$ as positive if $\boldsymbol{z}_i^\top \boldsymbol{w} + b \geq 0$ and vise versa.

In Proposition 3.1 and Corollary 3.2 below, we derive the distribution of node representations $\{\boldsymbol{z}_1, \cdots, \boldsymbol{z}_N\}$, and give the analytical form of the optimal parameters and expected accuracy.

**Proposition 3.1.** *For graphs generated by CSBM($\boldsymbol{\mu}_+, \boldsymbol{\mu}_-, d, h$), the node representation $\boldsymbol{z}_i$ of node $v_i \in \mathbb{C}_+$ generated by a single-layer GCN follows a Gaussian distribution of*

$$\boldsymbol{z}_i \sim \mathcal{N}\left((1 + \gamma h_i)\boldsymbol{\mu}_+ + \gamma(1 - h_i)\boldsymbol{\mu}_-, \left(1 + \frac{\gamma^2}{d_i}\right)\boldsymbol{I}\right), \tag{2}$$

*where $d_i$ is the degree of node $v_i$, and $h_i$ is the homophily of node $v_i$ defined in Eq. (1). Similar results hold for $v_i \in \mathbb{C}_-$ after swapping $\boldsymbol{\mu}_+$ and $\boldsymbol{\mu}_-$.*

**Corollary 3.2.** *When $\boldsymbol{\mu}_+ = \boldsymbol{\mu}, \boldsymbol{\mu}_- = -\boldsymbol{\mu}$, and all nodes have the same homophily $h = \frac{p}{p+q}$ and degree $d = \frac{N(p+q)}{2}$, the classifier maximizes the expected accuracy when $\boldsymbol{w} = \text{sign}(1 + \gamma(2h-1)) \cdot \frac{\boldsymbol{\mu}}{\|\boldsymbol{\mu}\|_2}$ and $b = 0$. It gives a linear decision boundary of $\{\boldsymbol{z} : \boldsymbol{z}^\top \boldsymbol{w} = 0\}$ and the expected accuracy*

$$Acc = \Phi\left(\sqrt{\frac{d}{d + \gamma^2}} \cdot |1 + \gamma(2h-1)| \cdot \|\boldsymbol{\mu}\|_2\right), \tag{3}$$

*where $\Phi$ is the CDF of the standard normal distribution.*

To analyze the distinct impact patterns of attribute shifts and structure shifts, we decompose the accuracy gap of GNNs between the source graph and the target graph into two parts as follows,

$$\underbrace{\mathrm{Acc}_S(g_S \circ f_S) - \mathrm{Acc}_T(g_S \circ f_S)}_{\text{total accuracy gap}} = \underbrace{\mathrm{Acc}_S(g_S \circ f_S) - \sup_{g_T} \mathrm{Acc}_T(g_T \circ f_S)}_{\text{representation degradation } \Delta_f} + \underbrace{\sup_{g_T} \mathrm{Acc}_T(g_T \circ f_S) - \mathrm{Acc}_T(g_S \circ f_S)}_{\text{classifier bias } \Delta_g},$$

where $\mathrm{Acc}_S, \mathrm{Acc}_T$ denote the accuracies on the source and target graphs, respectively. $\sup_{g_T} \mathrm{Acc}_T(g_T \circ f_S)$ is the highest accuracy that a GNN can achieve on the target graph when the featurizer $f_S$ is frozen and the classifier $g_T$ is allowed to adapt. Using this accuracy as a pivot, the accuracy gap is decomposed into representation degradation and classifier bias. A visualized illustration is shown in Figure 7.

- *Representation degradation* $\Delta_f$ quantifies the performance gap attributed to the suboptimality of the source featurizer $f_S$. Intuitively, this term measures the minimal performance gap between the source and target graphs that the GNN model can achieve by tuning the classifier $g_T$.
- *Classifier bias* $\Delta_g$ quantifies the performance gap attributed to the suboptimality of the source classifier $g_S$. Intuitively, this term measures the part of performance gap on the target graph the GNN model can reduce by tuning the classifier $g_T$.

**Proposition 3.3** (Impacts of attribute shifts). *When training a single-layer GCN on a source graph of CSBM($\boldsymbol{\mu}, -\boldsymbol{\mu}, d, h$), while testing it on a target graph of CSBM($\boldsymbol{\mu} + \Delta\boldsymbol{\mu}, -\boldsymbol{\mu} + \Delta\boldsymbol{\mu}, d, h$) with $\|\Delta\boldsymbol{\mu}\|_2 < |\frac{1+\gamma(2h-1)}{1+\gamma}| \cdot \|\boldsymbol{\mu}\|_2$, we have*

$$\Delta_f = 0, \qquad \Delta_g = \Theta(\|\Delta\boldsymbol{\mu}\|_2^2), \tag{4}$$

*where $\Theta$ indicates the same order, i.e., a function $l(x) = \Theta(x) \Leftrightarrow$ there exists positive constants $C_1, C_2$, s.t. $C_1 \leq \frac{l(x)}{x} \leq C_2$ for all $x$ in its range. It implies that the performance gap under attribute shifts mainly attributes to the classifier bias.*

**Proposition 3.4** (Impacts of structure shifts). *When training a single-layer GCN on a source graph of CSBM($\boldsymbol{\mu}, -\boldsymbol{\mu}, d_S, h_S$), while testing it on a target graph of CSBM($\boldsymbol{\mu}, -\boldsymbol{\mu}, d_T, h_T$), where $1 \leq d_T = d_S - \Delta d < d_S$ and $\frac{1}{2} < h_T = h_S - \Delta h < h_S$, if $\gamma > 0$, we have*

$$\Delta_f = \Theta(\Delta h + \Delta d), \qquad \Delta_g = 0, \tag{5}$$

*which implies that the performance gap under structure shifts mainly attributes to the representation degradation.*

Propositions 3.3 and 3.4 imply that attribute shifts and structure shifts impact the accuracy of GNN differently. Specifically, attribute shifts impact the decision boundary of the classifier, while structure shifts significantly degrade the node representation quality. These propositions also match with our empirical findings in Figure 2 and Figure 10 (in Appendix C.1). Since generic TTA methods (Wang et al., 2021; Iwasawa & Matsuo, 2021; Zhang et al., 2023) usually rely on the representation quality and refine the decision boundary, their effectiveness is limited under structure shifts.

### 3.3 ADAPTING HOP-AGGREGATION PARAMETERS TO RESTORE REPRESENTATIONS

To mitigate the representation degradation caused by structure shifts, it becomes essential to adjust the featurizer of GNNs. In the following Proposition 3.5, we demonstrate that the degraded node representations due to structure shifts can be restored by adapting $\gamma$, the hop-aggregation parameter. This is because $\gamma$ determines the way to combine a node's own attributes with its neighbors in GNNs. Notice that although our theory mainly focuses on single-layer GCNs, a wide range of GNN models possess similar parameters for adaptation, e.g., the general PageRank parameters in GPRGNN (Chien et al., 2021), teleport probability in APPNP (Klicpera et al., 2019), layer aggregation in JKNet (Xu et al., 2018). We extend our analysis to multi-hop GCNs in Appendix A.7.

**Proposition 3.5** (Adapting $\gamma$). *Under the same learning setting as Proposition 3.4, adapting the source $\gamma_S$ to the optimal $\gamma_T = d_T(2h_T - 1)$ on the target graph can alleviate the representation degradation and improve the target classification accuracy by $\Theta((\Delta h)^2 + (\Delta d)^2)$.*

Proposition 3.5 indicates that the optimal $\gamma$ depends on both node degree $d$ and homophily $h$. For instance, consider a source graph with $h_S = 1$ and $d_S = 10$. In this case, the optimal featurizer assigns equal weight to the node itself and each of its neighbors, resulting in optimal $\gamma_S = 10$.

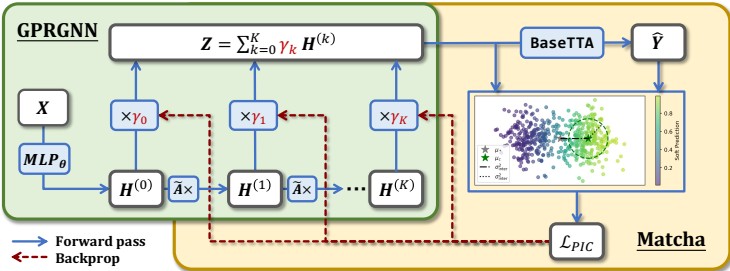

Figure 3: Our proposed framework of `Matcha` (when combined with GPRGNN)

However, when the target graph's degree remains unchanged but the homophily decreases to $h_T = 0.5$, where each node's neighbors are equally likely to be positive or negative, the neighbors no longer provide reliable information for node classification, leading to an optimal $\gamma_T = 0$. Similarly, when the homophily remains the same, but the target graph's degree is reduced to $d_T = 1$, $\gamma_S$ overemphasizes the neighbor's representation by placing excessive weight on it, whereas the optimal $\gamma_T$ in this case would be 1. A visualization of these examples are given in Appendix A.2.

## 4 PROPOSED FRAMEWORK

So far, we have found that adjusting hop-aggregation parameters can address the issue of node representation degradation caused by structure shifts. However, translating this theoretical insight into a practical algorithm still faces two challenges:

- In the absence of labels, how to update hop-aggregation parameters to handle structure shifts?
- How to ensure that our proposed algorithm is compatible with existing TTA algorithms (`BaseTTA`) in order to simultaneously address the co-existence of structure and attribute shifts?

In this section, we propose `Matcha`, including a novel prediction-informed clustering loss to encourage high-quality node representations, and an adaptation framework compatible with a wide range of TTA algorithms. Figure 3 gives a general framework.

To adapt to graphs with different degree distributions and homophily, `Matcha` uses GNNs that are capable of adaptively integrating multi-hop information, e.g., GPRGNN (Chien et al., 2021), APPNP (Klicpera et al., 2019), JKNet (Xu et al., 2018), etc. Specifically, we illustrate our framework using GPRGNN as a representative case. Notably, our framework's applicability extends beyond this example, as demonstrated by the experimental results presented in Appendix C.10, showcasing its versatility across various network architectures.

**GPRGNN.** The featurizer of GPRGNN is an MLP followed by a general pagerank module. We denote the parameters for MLP as $\boldsymbol{\theta}$, and the parameters for the general pagerank module as $\boldsymbol{\gamma} = [\gamma_0, \cdots, \gamma_K] \in \mathbb{R}^{K+1}$. The node representation of GPRGNN can be computed as $\boldsymbol{Z} = \sum_{k=0}^{K} \gamma_k \boldsymbol{H}^{(k)}$, where $\boldsymbol{H}^{(0)} = \text{MLP}_{\boldsymbol{\theta}}(\boldsymbol{X})$, $\boldsymbol{H}^{(k)} = \tilde{\boldsymbol{A}}^k \boldsymbol{H}^{(0)}, \forall k = 1, ..., K$ are the 0-hop and $k$-hop representations, $\tilde{\boldsymbol{A}}$ is the normalized adjacency matrix. A linear layer with weight $\boldsymbol{w}$ following the featurizer serves as the classifier.

### 4.1 PREDICTION-INFORMED CLUSTERING LOSS

This subsection introduces how `Matcha` updates the hop-aggregation parameters without labels. Previous TTA methods (Liang et al., 2020; Wang et al., 2021; Zhang et al., 2022; Bao et al., 2023) mainly adopt the entropy as a surrogate loss, as it measures the prediction uncertainty. However, we find that entropy minimization has limited effectiveness in improving representation quality (see Figure 4 and Table 7). Entropy is sensitive to the scale of logits rather than representation quality, often leading to trivial solutions. For instance, for a linear classifier, simply scaling up all the node representations can cause the entropy loss to approach zero, without improving the separability of the node representations between different classes. To address this issue, we propose the prediction-informed clustering (PIC) loss, which can better reflect the quality of node representation under structure shifts. Minimizing the PIC loss encourages the representations of nodes from different classes to be more distinct and less overlapping.

Let $\boldsymbol{Z} = [\boldsymbol{z}_1, \cdots, \boldsymbol{z}_N]^\top \in \mathbb{R}^{N \times D}$ denote the representation matrix and $\hat{Y} \in \mathbb{R}_+^{N \times C}$ denote the prediction of `BaseTTA` subject to $\sum_{c=1}^{C} \hat{Y}_{i,c} = 1$, where $N$ is the number of nodes, $D$ is the dimension of the node representations and $C$ is the number of classes. We first compute $\boldsymbol{\mu}_c$ as the centroid representation of each (pseudo-)class $c$, and $\boldsymbol{\mu}_*$ as the centroid representation for all nodes,

$$\boldsymbol{\mu}_c = \frac{\sum_{i=1}^{N} \hat{Y}_{i,c} \boldsymbol{z}_i}{\sum_{i=1}^{N} \hat{Y}_{i,c}}, \quad \forall c = 1, \cdots, C, \qquad \boldsymbol{\mu}_* = \frac{1}{N} \sum_{i=1}^{N} \boldsymbol{z}_i. \tag{6}$$

We further define the intra-class variance $\sigma_{\text{intra}}^2$ and inter-class variance $\sigma_{\text{inter}}^2$ as:

$$\sigma_{\text{intra}}^2 = \sum_{i=1}^{N} \sum_{c=1}^{C} \hat{Y}_{i,c} \|\boldsymbol{z}_i - \boldsymbol{\mu}_c\|_2^2, \qquad \sigma_{\text{inter}}^2 = \sum_{c=1}^{C} \left( \sum_{i=1}^{N} \hat{Y}_{i,c} \right) \|\boldsymbol{\mu}_c - \boldsymbol{\mu}_*\|_2^2. \tag{7}$$

To obtain discriminative representations, it is natural to expect small intra-class variance $\sigma_{\text{intra}}^2$, i.e., nodes with the same label are clustered together, and high inter-class variance $\sigma_{\text{inter}}^2$, i.e., different classes are separated. Therefore, we propose the PIC loss as follows:

$$\mathcal{L}_{\text{PIC}} = \frac{\sigma_{\text{intra}}^2}{\sigma_{\text{intra}}^2 + \sigma_{\text{inter}}^2} = \frac{\sigma_{\text{intra}}^2}{\sigma^2}, \tag{8}$$

where $\sigma^2$ can be simplified as $\sigma^2 = \sigma_{\text{intra}}^2 + \sigma_{\text{inter}}^2 = \sum_{i=1}^{N} \|\boldsymbol{z}_i - \boldsymbol{\mu}_*\|_2^2$ (proof in Appendix A.8).

It should be noted that although the form of PIC loss seems not reusing the adjacency matrix $\boldsymbol{A}$, it still evaluates the suitability of the current hop-aggregation parameters for the graph structure through the distribution of the representation $\boldsymbol{Z}$. As shown in Figure 4 and Proposition 3.4, structure shifts cause node representations to overlap more, leading to a smaller $\sigma_{\text{inter}}^2 / \sigma_{\text{intra}}^2$ and a larger PIC loss. Alternatively, some algorithms, like SOGA (Mao et al., 2024), incorporate edge information by promoting connected nodes to share the same label. These designs implicitly assume of homophilic graph, limiting their applicability. As a result, SOGA performs poorly on heterophilic target graphs, as seen in Table 1. In contrast, our PIC loss directly targets GNN-encoded node representations, allowing it to generalize across different graph structures, whether homophilic or heterophilic.

By minimizing the PIC loss, we reduce intra-class variance while maximizing inter-class variance. Importantly, the ratio form of the PIC loss reduces sensitivity to the scale of representations; as the norm increases, the loss does not converge to zero, thus avoiding trivial solutions. It is also worth noting that the proposed PIC loss differs from the Fisher score (Gu et al., 2012) in two key aspects: First, PIC loss operates on model predictions, while Fisher score relies on true labels, making Fisher inapplicable in our setting where labels are unavailable. Second, PIC loss uses soft predictions for variance computation, which aids in the convergence of `Matcha`, whereas the Fisher score uses hard labels, which can lead to poor convergence due to the unbounded Lipschitz constant, as we show in Theorem 4.1. We also provide an example in Appendix C.2 showing that `Matcha` with PIC loss improves accuracy even when initial predictions are highly noisy.

## 4.2 Integration of generic TTA methods

This subsection introduces how `Matcha` integrates the adaptation of hop-aggregation parameters with existing TTA algorithms to simultaneously address the co-existence of structure and attribute shifts. Our approach is motivated by the complementary nature of adapting the hop-aggregation parameter and existing generic TTA methods. While the adapted hop-aggregation parameter effectively manages structure shifts, generic TTA methods handle attribute shifts in various ways. Consequently,

---

**Algorithm 1** `Matcha`

---

**`Matcha`** (target graph $\mathcal{T}$, featurizer $f_{\boldsymbol{\theta},\boldsymbol{\gamma}}$, classifier $g_{\boldsymbol{w}}$, baseline TTA method `BaseTTA`)

1: **for** epoch $t = 1$ to $T$ **do**
2:     Apply generic TTA:
        $\hat{Y} \leftarrow \text{BaseTTA}(\mathcal{T}, f_{\boldsymbol{\theta},\boldsymbol{\gamma}}, g_{\boldsymbol{w}})$
3:     Update hop-aggregation parameters:
        $\boldsymbol{\gamma} \leftarrow \boldsymbol{\gamma} - \eta \nabla_{\boldsymbol{\gamma}} \mathcal{L}(\mathcal{T}, f_{\boldsymbol{\theta},\boldsymbol{\gamma}}, g_{\boldsymbol{w}}, \hat{Y})$
4: **return** $\hat{Y} \leftarrow \text{BaseTTA}(\mathcal{T}, f_{\boldsymbol{\theta},\boldsymbol{\gamma}}, g_{\boldsymbol{w}})$

---

we design a simple yet effective framework that seamlessly integrates the adaptation of hop-aggregation parameter with a broad range of existing generic TTA techniques.

Our proposed `Matcha` framework is illustrated in Algorithm 1. Given a pre-trained source GNN model $f_{\boldsymbol{\theta},\boldsymbol{\gamma}} \circ g_{\boldsymbol{w}}$ and the target graph $\mathcal{T}$, we first employ the baseline TTA method, named `BaseTTA`,

to produce the soft prediction $\hat{Y} \in \mathbb{R}_+^{N \times C}$ as pseudo-classes, where $\sum_{c=1}^C \hat{Y}_{i,c} = 1$. Equipped with pseudo-classes, the hop-aggregation parameters $\gamma$ is adapted by minimizing the PIC loss as described in Subsection 4.1. Intuitively, the predictions of `BaseTTA` are crucial for identifying pseudo-classes to cluster representations, and in return, better representations enhance the prediction accuracy of `BaseTTA`. Such synergy between representation quality and prediction accuracy mutually reinforces each other during the adaptation process, leading to much more effective outcomes. It is worth noting that `Matcha` is a plug-and-play method that can seamlessly integrate with various TTA algorithms, including Tent (Wang et al., 2021), T3A (Iwasawa & Matsuo, 2021), and AdaNPC (Zhang et al., 2023).

**Computational complexity.** For each epoch, the computational complexity of the PIC loss is $\mathcal{O}(NCD)$, linear to the number of nodes. Compared to SOGA (Mao et al., 2024), which has quadratic complexity from comparing every node pair, PIC loss enjoys greater scalability to the graph size. For the whole `Matcha` framework, it inevitably introduces additional computational overhead, which depends on both the GNN architecture and the baseline TTA algorithm. However, in practice, the additional computational cost is generally minimal since intermediate results (e.g. $\{H^{(k)}\}_{k=0}^K$) can be cached and reused. We empirically evaluate the efficiency of `Matcha` in Subsection 5.3, Appendix C.8, and the scalability in Appendix C.9.

**Convergence analysis.** Finally, we analyze the convergence property of `Matcha` in Theorem 4.1 below. The formal theorem and complete proofs can be found in Appendix B.

**Theorem 4.1** (Convergence of `Matcha`). *Let $M = [vec(H^{(0)}), \cdots, vec(H^{(K)})] \in \mathbb{R}^{ND \times (K+1)}$ denote the concatenation of 0-hop to $K$-hop node representations. Given a base TTA algorithm, if (1) the prediction $\hat{Y}$ is $L$-Lipschitz w.r.t. the (aggregated) node representation $Z$, and (2) the loss function is $\beta$-smooth w.r.t. $Z$, after $T$ steps of gradient descent with step size $\eta = \frac{1}{\beta \|M\|_2^2}$, we have*

$$\frac{1}{T} \sum_{t=0}^T \left\| \nabla_\gamma \mathcal{L}(\gamma^{(t)}) \right\|_2^2 \leq 2 \frac{\beta \|M\|_2^2}{T} \mathcal{L}(\gamma^{(0)}) + CL^2 \|M\|_2^2, \tag{9}$$

*where $C$ is a constant.*

Theorem 4.1 shows that `Matcha` is guaranteed to converge to a flat region with small gradients, with convergence rate $\frac{1}{T}$ and error rate $\propto L^2$. Essentially, the convergence of `Matcha` depends on the sensitivity of the `BaseTTA` algorithm. Intuitively, if `BaseTTA` has large Lipschitz constant $L$, it is likely to make completely different predictions in each epoch, and thus hindering the convergence of `Matcha`. However, in general cases, $L$ is upper bounded. We give theoretical verification in Lemma B.9 under ERM, and further empirically verify the convergence of `Matcha` in Figure 6.

## 5 EXPERIMENTS

We conduct extensive experiments on synthetic and real-world datasets to evaluate our proposed `Matcha` from the following aspects:

- **RQ1**: How can `Matcha` empower TTA algorithms and handle various structure shifts on graphs?
- **RQ2**: To what extent can `Matcha` restore the representation quality better than other methods?

### 5.1 MATCHA HANDLES VARIOUS STRUCTURE SHIFTS (RQ1)

**Experiment setup.** We first adopt CSBM (Deshpande et al., 2018) to generate synthetic graphs with controlled structure and attribute shifts. We consider a hybrid of attribute shift, homophily shift and degree shift. For homophily shift, we generate a homophily graph with $h = 0.8$ and a heterophily graph with $h = 0.2$. For degree shift, we generate a high-degree graph with $d = 10$ and a low-degree graph with $d = 2$. For attribute shift, we transform the class centers $\mu_+, \mu_-$ on the target graph. For real-world datasets, we adopt Syn-Cora (Zhu et al., 2020), Syn-Products (Zhu et al., 2020), Twitch-E (Rozemberczki et al., 2021), and OGB-Arxiv (Hu et al., 2020). For Syn-Cora and Syn-Products, we use $h = 0.8$ as the source graph and $h = 0.2$ has the target graph. For Twitch-E and OGB-Arxiv, we delete a subset of homophilic edges in the target graph to inject both degree and homophily shifts. The detailed dataset statistics are provided in Appendix D.1.

We adopt GPRGNN (Chien et al., 2021) as the backbone model for the main experiments. We also provide results on other backbone models, including APPNP (Klicpera et al., 2019), JKNet (Xu

Table 1: Accuracy (mean $\pm$ s.d. %) on CSBM with structure shifts and attribute shifts.

| Method | Homophily shift | | Degree shift | | Attribute + homophily shift | | Attribute + degree shift | |
|---|---|---|---|---|---|---|---|---|
| | homo → hetero | hetero → homo | high → low | low → high | homo → hetero | hetero → homo | high → low | low → high |
| ERM | 73.62 ± 0.44 | 76.72 ± 0.89 | 86.47 ± 0.38 | 92.92 ± 0.43 | 61.06 ± 1.67 | 72.61 ± 0.38 | 77.63 ± 1.13 | 73.60 ± 3.53 |
| + Matcha | 89.71 ± 0.27 | 90.68 ± 0.26 | 88.55 ± 0.44 | 93.78 ± 0.74 | 85.34 ± 4.68 | 74.70 ± 0.99 | 78.29 ± 1.41 | 73.86 ± 4.20 |
| T3A | 73.85 ± 0.24 | 76.68 ± 1.08 | 86.52 ± 0.44 | 92.94 ± 0.37 | 65.77 ± 2.11 | 72.92 ± 0.90 | 80.89 ± 1.28 | 81.94 ± 3.24 |
| + Matcha | 90.40 ± 0.11 | 90.50 ± 0.24 | 88.42 ± 0.60 | 93.83 ± 0.41 | 88.49 ± 0.58 | 79.34 ± 1.85 | 81.82 ± 1.36 | 82.12 ± 4.03 |
| Tent | 74.64 ± 0.38 | 79.40 ± 0.57 | 86.49 ± 0.50 | 92.84 ± 0.18 | 74.42 ± 0.41 | 79.57 ± 0.40 | 86.05 ± 0.33 | 93.06 ± 0.24 |
| + Matcha | 89.93 ± 0.16 | **91.26 ± 0.08** | **89.20 ± 0.20** | **94.88 ± 0.09** | **90.12 ± 0.07** | **91.15 ± 0.20** | **87.76 ± 0.16** | **95.04 ± 0.06** |
| AdaNPC | 76.03 ± 0.46 | 81.66 ± 0.17 | 86.92 ± 0.38 | 91.15 ± 0.39 | 63.96 ± 1.31 | 76.33 ± 0.71 | 77.69 ± 0.91 | 76.24 ± 3.06 |
| + Matcha | 90.03 ± 0.33 | 90.36 ± 0.67 | 88.49 ± 0.31 | 92.84 ± 0.57 | 85.81 ± 0.30 | 77.63 ± 1.55 | 78.41 ± 1.03 | 76.31 ± 3.68 |
| GTrans | 74.01 ± 0.44 | 77.28 ± 0.56 | 86.58 ± 0.11 | 92.74 ± 0.13 | 71.60 ± 0.60 | 74.45 ± 0.42 | 83.21 ± 0.25 | 89.40 ± 0.62 |
| + Matcha | 89.47 ± 0.20 | 90.31 ± 0.31 | 87.88 ± 0.77 | 93.23 ± 0.52 | 88.88 ± 0.38 | 76.87 ± 0.66 | 83.41 ± 0.16 | 89.98 ± 0.93 |
| SOGA | 74.33 ± 0.18 | 83.99 ± 0.35 | 86.69 ± 0.37 | 93.06 ± 0.21 | 70.45 ± 1.71 | 76.41 ± 0.79 | 81.31 ± 1.03 | 88.32 ± 1.94 |
| + Matcha | 89.92 ± 0.26 | 90.69 ± 0.27 | 88.83 ± 0.32 | 94.49 ± 0.23 | 88.92 ± 0.28 | 90.14 ± 0.33 | 87.11 ± 0.28 | 93.38 ± 1.06 |
| GraphPatcher | 79.14 ± 0.62 | 82.14 ± 1.11 | 87.87 ± 0.18 | 93.64 ± 0.45 | 64.16 ± 3.49 | 76.98 ± 1.04 | 76.99 ± 1.43 | 73.31 ± 4.48 |
| + Matcha | **91.28 ± 0.28** | 90.66 ± 0.15 | 88.01 ± 0.18 | 93.88 ± 0.69 | 89.99 ± 0.41 | 87.94 ± 0.39 | 78.43 ± 1.84 | 77.86 ± 4.14 |

Table 2: Accuracy on real-world datasets.

| Method | Syn-Cora | Syn-Products | Twitch-E | OGB-Arxiv |
|---|---|---|---|---|
| ERM | 65.67 ± 0.35 | 37.80 ± 2.61 | 56.20 ± 0.63 | 41.06 ± 0.33 |
| + Matcha | 78.96 ± 1.08 | 69.75 ± 0.93 | 56.76 ± 0.22 | 41.74 ± 0.34 |
| T3A | 68.25 ± 1.10 | 47.59 ± 1.46 | 56.83 ± 0.22 | 38.17 ± 0.31 |
| + Matcha | 78.40 ± 1.04 | 69.81 ± 0.36 | 56.97 ± 0.28 | 38.56 ± 0.27 |
| Tent | 66.26 ± 0.38 | 29.14 ± 4.50 | 58.46 ± 0.37 | 34.48 ± 0.28 |
| + Matcha | 78.87 ± 1.07 | 68.45 ± 1.04 | **58.57 ± 0.42** | 35.20 ± 0.27 |
| AdaNPC | 67.34 ± 0.76 | 44.67 ± 1.53 | 55.43 ± 0.50 | 40.20 ± 0.35 |
| + Matcha | 77.45 ± 0.62 | 71.66 ± 0.81 | 56.35 ± 0.27 | 40.58 ± 0.35 |
| GTrans | 68.60 ± 0.32 | 43.89 ± 1.75 | 56.24 ± 0.41 | 41.28 ± 0.31 |
| + Matcha | **83.49 ± 0.78** | **71.75 ± 0.65** | 56.75 ± 0.40 | 41.81 ± 0.31 |
| SOGA | 67.16 ± 0.72 | 40.96 ± 2.87 | 56.12 ± 0.30 | 41.23 ± 0.34 |
| + Matcha | 79.03 ± 1.10 | 70.13 ± 0.86 | 56.62 ± 0.17 | 41.78 ± 0.34 |
| GraphPatcher | 63.01 ± 2.29 | 36.94 ± 1.50 | 57.05 ± 0.59 | 41.27 ± 0.87 |
| + Matcha | 80.99 ± 0.50 | 69.39 ± 1.29 | 57.41 ± 0.53 | **41.83 ± 0.90** |

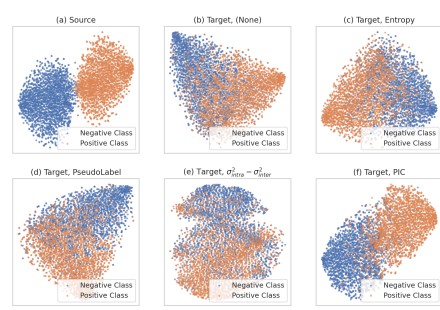

Figure 4: T-SNE visualization of node representations on CSBM homo → hetero.

et al., 2018), and GCNII (Chen et al., 2020) in Appendix C.10. Details on model architectures are provided in Appendix D.2. We run each experiment five times with different random seeds and report the mean accuracy and standard deviation.

**Baselines.** We consider two groups of base TTA methods, including: (1) generic TTA methods: T3A (Iwasawa & Matsuo, 2021), Tent (Wang et al., 2021), and AdaNPC (Zhang et al., 2023), and (2) graph TTA methods: GTrans (Jin et al., 2023), SOGA (Mao et al., 2024) and GraphPatcher (Ju et al., 2023). To ensure a fair comparison, we focus on TTA algorithms in the same setting, which adapt a pre-trained model to a target graph without re-accessing the source graph. We adopt Empirical Risk Minimization (ERM) to pre-train the model on the source graph without adaptation. We use the node classification accuracy on the target graph to evaluate the model performance.

**Main Results.** The experimental results on the CSBM dataset are shown in Table 1. Under various shifts, the proposed `Matcha` consistently enhances the performance of base TTA methods. Specifically, compared to directly using the pre-trained model without adaptation (ERM), adopting `Matcha` (ERM+`Matcha`) could significantly improve model performance, with up to 24.28% improvements. Compared with other baseline methods, `Matcha` achieves the best performance in most cases, with up to 21.38% improvements. Besides, since `Matcha` is compatible and complementary with the baseline TTA methods, we also compare the performance of baseline methods with and without `Matcha`. As the results show, `Matcha` could further boost the performance of TTA baselines by up to 22.72%.

For real-world datasets, the experimental results are shown in Table 2. Compared with ERM, `Matcha` could significantly improve the model performance by up to 31.95%. Compared with other baseline methods, `Matcha` achieves comparable performance on Twitch-E, and significant improvements on Syn-Cora, Syn-Products and OGB-Arxiv, with up to 40.61% outperformance. When integrated with other TTA methods, `Matcha` can further enhance the performance by up to 39.31%. The significant outperformance verifies the effectiveness of the proposed `Matcha`.

**Additional experiments.** We also demonstrate that `Matcha` exhibits robustness against (1) *structure shifts of varying levels*, (2) *evolving target graph*, and (3) *additional adversarial shifts* in Appendix C.3, C.4, and C.5, respectively.

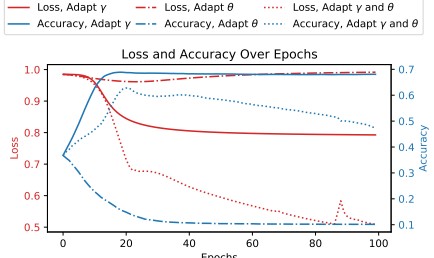 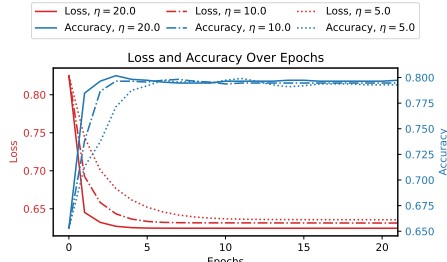

Figure 5: Ablation study on Syn-Products with different sets of parameters to adapt.

Figure 6: Convergence of `Matcha` on Syn-Cora with different learning rates $\eta$.

## 5.2 MATCHA RESTORES THE REPRESENTATION QUALITY (RQ2)

Besides the superior performance of `Matcha`, we are also interested in whether `Matcha` successfully restores the quality of node representations under structure shifts. To explore this, we visualize the learned node representations on 2-class CSBM graphs in Figure 4. Although the pre-trained model generates high-quality node representations (Figure 4(a)), node representations degrades dramatically when directly deploying the source model to the target graph without adaptation (Figure 4(b)). With our proposed PIC loss, `Matcha` successfully restores the representation quality with a clear cluster structure (Figure 4(f)). Moreover, compared to other common surrogate losses (entropy, pseudo-label), PIC loss results in significantly better representations.

## 5.3 MORE DISCUSSIONS

**Ablation study.** While `Matcha` adapts only the hop-aggregation parameters $\gamma$ to improve representation quality, other strategies exist, such as adapting the MLP parameters $\theta$ or both $\gamma$ and $\theta$ together. As shown in Figure 5, adapting only $\theta$ fails to significantly reduce the PIC loss or improve accuracy. Adapting both $\gamma$ and $\theta$ minimizes the PIC loss but leads to model forgetting, causing an initial accuracy increase followed by a decline. In contrast, adapting only $\gamma$ results in smooth loss convergence and stable accuracy, demonstrating that `Matcha` effectively adapts to structure shifts without forgetting source graph information. We also compare our proposed PIC loss to other surrogate losses in Appendix C.6. Our PIC loss has better performance under four structure shifts.

**Hyperparameter sensitivity.** `Matcha` only introduces two hyperparameters including the learning rate $\eta$ and the number of epochs $T$. In Figure 6, we explore different combinations of them. We observe that `Matcha` converges smoothly in just a few epochs, and the final loss and accuracy are quite robust to various choices of the learning rate. Additionally, as discussed in Appendix C.7, we examine the effect of the dimension of hop-aggregation parameters $K$ on `Matcha`, and find that it consistently provides stable accuracy gains across a wide range of $K$ values.

**Computational efficiency.** We quantify the additional computation time introduced by `Matcha` during the test-time. Compared to the standard inference time, `Matcha` *only adds an extra 11.9% in computation time for each epoch of adaptation*. In comparison, GTrans and SOGA adds 486% and 247% in computation time. `Matcha` enjoys great efficiency resulting from only updating the hop-aggregation parameters and efficient loss design. Please refer to Appendix C.8 for more details.

**Compatibility to more GNN architectures.** Besides GPRGNN, `Matcha` is compatible with various GNN architectures, e.g., JKNet (Xu et al., 2018), APPNP (Klicpera et al., 2019), and GCNII (Chen et al., 2020). In Appendix C.10, we test the performance of `Matcha` with these networks on Syn-Cora. `Matcha` consistently improves the accuracy.

## 6 CONCLUSION

In this paper, we explore why generic TTA algorithms perform poorly under structure shifts. Theoretical analysis reveals that attribute structure shifts on graphs bear distinct impact patterns on the GNN performance, where the attribute shifts introduce classifier bias while the structure shifts degrade the node representation quality. Guided by this insight, we propose `Matcha`, a plug-and-play TTA framework that restores the node representation quality with convergence guarantee. Extensive experiments consistently and significantly demonstrate the effectiveness of `Matcha`.

ACKNOWLEDGMENT

This work is supported by National Science Foundation under Award No. IIS-2416070, and the U.S. Department of Homeland Security under Grant Award Number 17STQAC00001-08-00. The views and conclusions are those of the authors and should not be interpreted as representing the official policies of the funding agencies or the government.

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

APPENDIX CONTENTS

# A  THEORETICAL ANALYSIS

## A.1  ILLUSTRATION OF REPRESENTATION DEGRADATION AND CLASSIFIER BIAS

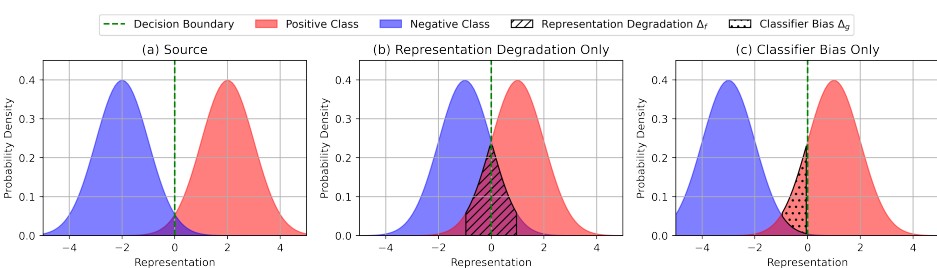

Figure 7: An example of representation degradation and classifier bias. (b) Representation degradation blurs the boundary between two classes and increases their overlap. (c) Classifier bias translates the representation and makes the decision boundary sub-optimal.

Figure 7 above visualizes representation degradation and classifier bias.

- Figure 7(c): Under classifier bias, the representations are shifted to the left, making the decision boundary sub-optimal. However, by refining the decision boundary, the accuracy can be fully recovered.

- Figure 7(b): Under representation degradation, however, even if we refine the decision boundary, the accuracy cannot be recovered without changing the node representations.

Moreover, comparing Figure 7 with Figure 10, we can clearly conclude that attribute shifts mainly introduce classifier bias, while structure shift mainly introduce representation degradation.

## A.2    ILLUSTRATION OF ADAPTING $\gamma$

In the end of subsection 3.3, we provide two examples to intuitively illustrate why the hop-aggregation parameter $\gamma$ should be adjusted based on the target graph's degree $d_T$ and homophily $h_T$. To further demonstrate the effect of adapting $\gamma$, we visualize these examples:

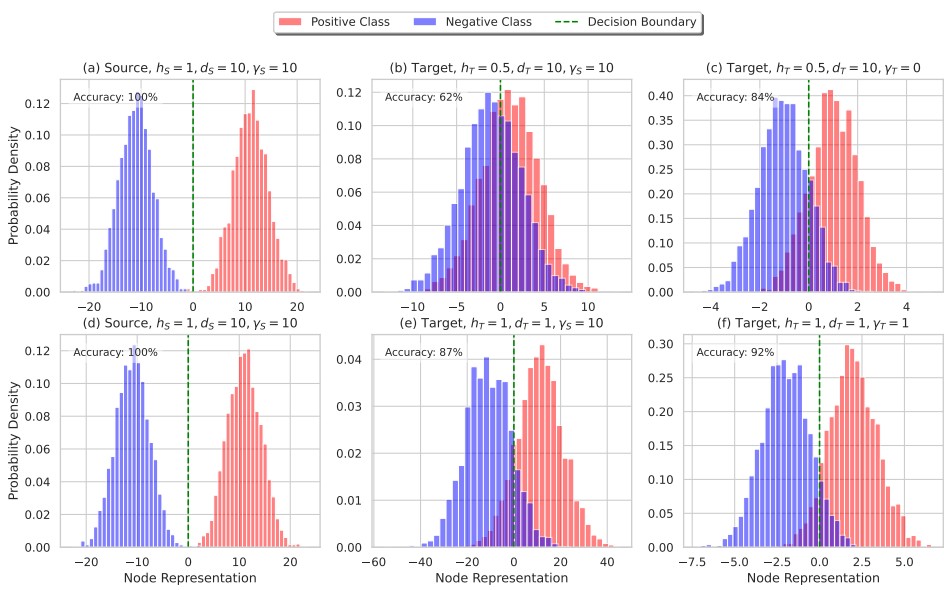

Figure 8: Visualization of effect of structure shifts and adaptation of $\gamma$.

- **Source Graph**: We consider a source graph with $\mu_+ = 1, \mu_- = -1, h_S = 1$, and $d_S = 10$. In this case, the optimal featurizer assigns equal weight to the node itself and each of its neighbors, resulting in the optimal $\gamma_S = 10$. Figure 8(a) and (d) show the distribution of node representations on the source graph, where the two classes are well-separated.

- **Homophily Shift**: In Figure 8(b), the degree remains unchanged, but the homophily $h_T$ decreases to 0.5, meaning each node's neighbors are equally likely to belong to either class. The neighbors no longer provide reliable information for classification, introducing noise and reducing accuracy to 62%. However, in Figure 8 (c), after adjusting $\gamma_T$ to the optimal $d_T(2h_T - 1) = 0$, the accuracy improves significantly to 84%.

- **Degree Shift**: In Figure 8(e), the homophily remains unchanged, but the degree $d_T$ decreases to 1. The original $\gamma$ overemphasizes the neighbors' representations, placing excessive weight on them, leading to an accuracy drop to 87%. By adjusting $\gamma_T$ to the optimal $d_T(2h_T - 1) = 1$ in Figure 8(f), the accuracy improves to 92%.

## A.3   PROOF OF PROPOSITION 3.1 AND COROLLARY 3.2

**Proposition 3.1.** *For graphs generated by CSBM($\boldsymbol{\mu}_+, \boldsymbol{\mu}_-, d, h$), the node representation $\boldsymbol{z}_i$ of node $v_i \in \mathbb{C}_+$ generated by a single-layer GCN follows a Gaussian distribution of*

$$\boldsymbol{z}_i \sim \mathcal{N}\left((1+\gamma h_i)\boldsymbol{\mu}_+ + \gamma(1-h_i)\boldsymbol{\mu}_-, \left(1+\frac{\gamma^2}{d_i}\right)\boldsymbol{I}\right), \tag{2}$$

*where $d_i$ is the degree of node $v_i$, and $h_i$ is the homophily of node $v_i$ defined in Eq. (1). Similar results hold for $v_i \in \mathbb{C}_-$ after swapping $\boldsymbol{\mu}_+$ and $\boldsymbol{\mu}_-$.*

*Proof.* For each node $v_i \in \mathbb{C}_+$, its representation is computed as

$$\boldsymbol{z}_i = \boldsymbol{x}_i + \gamma \cdot \frac{1}{d_i} \sum_{v_j \in \mathbb{N}(v_i)} \boldsymbol{x}_j$$

The linear combination of Gaussian distribution is still Gaussian. Among the $d_i = |\mathbb{N}(v_i)|$ neighbors of node $v_i$, there are $h_i d_i$ nodes from $\mathbb{C}_+$ and $(1-h_i)d_i$ nodes from $\mathbb{C}_-$. Therefore, the distribution of $\boldsymbol{z}_i$ is

$$\boldsymbol{z}_i \sim \mathcal{N}\left((1+\gamma h_i)\boldsymbol{\mu}_+ + \gamma(1-h_i)\boldsymbol{\mu}_-, \left(1+\frac{\gamma^2}{d_i}\right)\boldsymbol{I}\right)$$

Similarly, for each node $v_i \in \mathbb{C}_-$, the distribution of $\boldsymbol{z}_i$ is

$$\boldsymbol{z}_i \sim \mathcal{N}\left((1+\gamma h_i)\boldsymbol{\mu}_- + \gamma(1-h_i)\boldsymbol{\mu}_+, \left(1+\frac{\gamma^2}{d_i}\right)\boldsymbol{I}\right)$$

$\square$

*Remark* A.1.   When $\gamma \to \infty$, this proposition matches with the results in (Ma et al., 2022).[2]

---

[2]Notice that our notation is slightly different: we use the covariance matrix while they use the square root of it in the multivariate Gaussian distribution.

**Corollary 3.2.** *When $\boldsymbol{\mu}_+ = \boldsymbol{\mu}, \boldsymbol{\mu}_- = -\boldsymbol{\mu}$, and all nodes have the same homophily $h = \frac{p}{p+q}$ and degree $d = \frac{N(p+q)}{2}$, the classifier maximizes the expected accuracy when $\boldsymbol{w} = \text{sign}(1 + \gamma(2h-1)) \cdot \frac{\boldsymbol{\mu}}{\|\boldsymbol{\mu}\|_2}$ and $b = 0$. It gives a linear decision boundary of $\{\boldsymbol{z} : \boldsymbol{z}^\top \boldsymbol{w} = 0\}$ and the expected accuracy*

$$Acc = \Phi\left(\sqrt{\frac{d}{d + \gamma^2}} \cdot |1 + \gamma(2h-1)| \cdot \|\boldsymbol{\mu}\|_2\right), \tag{3}$$

*where $\Phi$ is the CDF of the standard normal distribution.*

*Proof.* Given $\boldsymbol{\mu}_+ = \boldsymbol{\mu}, \boldsymbol{\mu}_- = -\boldsymbol{\mu}$ and $d_i = d, h_i = h, \forall i$, we have

$$\boldsymbol{z}_i \sim \mathcal{N}\left((1 + \gamma(2h-1))y_i\boldsymbol{\mu}, \left(1 + \frac{\gamma^2}{d}\right)\boldsymbol{I}\right)$$

where $y_i \in \{\pm 1\}$ is the label of node $v_i$. Given two multivariate Gaussian distributions with identical isotropic covariance matrix, the optimal decision boundary that maximize the expected accuracy is the perpendicular bisector of the line segment connecting two distribution means, i.e.,

$$\{\boldsymbol{z} : \|\boldsymbol{z} - (1 + \gamma(2h-1))\boldsymbol{\mu}\|_2 = \|\boldsymbol{z} - (1 + \gamma(2h-1))(-\boldsymbol{\mu})\|_2\} = \{\boldsymbol{z} : \boldsymbol{z}^\top \boldsymbol{\mu} = 0\}$$

The corresponding classifier is:

$$\boldsymbol{w} = \text{sign}(1 + \gamma(2h-1)) \cdot \frac{\boldsymbol{\mu}}{\|\boldsymbol{\mu}\|_2}, \quad b = 0 \tag{10}$$

To compute the expected accuracy for classification, we consider the distribution of $\boldsymbol{z}_i^\top \boldsymbol{w} + b$.

$$\boldsymbol{z}_i^\top \boldsymbol{w} + b \sim \mathcal{N}\left(|1 + \gamma(2h-1)| \cdot y_i \cdot \|\boldsymbol{\mu}\|_2, 1 + \frac{\gamma^2}{d}\right) \tag{11}$$

We scale it to unit identity variance,

$$\sqrt{\frac{d}{d + \gamma^2}} \cdot (\boldsymbol{z}_i^\top \boldsymbol{w} + b) \sim \mathcal{N}\left(\sqrt{\frac{d}{d + \gamma^2}} \cdot |1 + \gamma(2h-1)| \cdot y_i \cdot \|\boldsymbol{\mu}\|_2, 1\right)$$

Therefore, the expected accuracy is

$$\text{Acc} = \Phi\left(\sqrt{\frac{d}{d + \gamma^2}} \cdot |1 + \gamma(2h-1)| \cdot \|\boldsymbol{\mu}\|_2\right) \tag{12}$$

where $\Phi$ is the CDF of the standard normal distribution. $\qquad\square$

A.4  PROOF OF PROPOSITION 3.3

**Proposition 3.3** (Impacts of attribute shifts). *When training a single-layer GCN on a source graph of CSBM($\boldsymbol{\mu}, -\boldsymbol{\mu}, d, h$), while testing it on a target graph of CSBM($\boldsymbol{\mu} + \Delta\boldsymbol{\mu}, -\boldsymbol{\mu} + \Delta\boldsymbol{\mu}, d, h$) with $\|\Delta\boldsymbol{\mu}\|_2 < |\frac{1+\gamma(2h-1)}{1+\gamma}| \cdot \|\boldsymbol{\mu}\|_2$, we have*

$$\Delta_f = 0, \qquad \Delta_g = \Theta(\|\Delta\boldsymbol{\mu}\|_2^2), \tag{4}$$

*where $\Theta$ indicates the same order, i.e., a function $l(x) = \Theta(x) \Leftrightarrow$ there exists positive constants $C_1, C_2$, s.t. $C_1 \leq \frac{l(x)}{x} \leq C_2$ for all $x$ in its range. It implies that the performance gap under attribute shifts mainly attributes to the classifier bias.*

*Proof.* We can reuse the results in Corollary 3.2 by setting $\boldsymbol{\mu}_+ = \boldsymbol{\mu} + \Delta\boldsymbol{\mu}$ and $\boldsymbol{\mu}_- = -\boldsymbol{\mu} + \Delta\boldsymbol{\mu}$. For each node $v_i$, we have

$$\boldsymbol{z}_i \sim \mathcal{N}\left((1 + \gamma(2h-1))y_i\boldsymbol{\mu} + (1+\gamma)\Delta\boldsymbol{\mu}, \left(1 + \frac{\gamma^2}{d}\right)\boldsymbol{I}\right)$$

Given the classifier in Corollary 3.2, we have

$$\sqrt{\frac{d}{d+\gamma^2}} \cdot (\boldsymbol{z}_i^\top \boldsymbol{w} + b)$$

$$\sim \begin{cases} \mathcal{N}\left(\sqrt{\frac{d}{d+\gamma^2}} \cdot |1 + \gamma(2h-1)| \cdot \|\boldsymbol{\mu}\|_2 + \right. \\ \left. \sqrt{\frac{d}{d+\gamma^2}} \cdot \text{sign}(1 + \gamma(2h-1)) \cdot (1+\gamma) \cdot \text{cos\_sim}(\boldsymbol{\mu}, \Delta\boldsymbol{\mu})\|\Delta\boldsymbol{\mu}\|_2, 1\right), \quad \forall v_i \in \mathbb{C}_+ \\ \mathcal{N}\left(-\sqrt{\frac{d}{d+\gamma^2}} \cdot |1 + \gamma(2h-1)| \cdot \|\boldsymbol{\mu}\|_2 + \right. \\ \left. \sqrt{\frac{d}{d+\gamma^2}} \cdot \text{sign}(1 + \gamma(2h-1)) \cdot (1+\gamma) \cdot \text{cos\_sim}(\boldsymbol{\mu}, \Delta\boldsymbol{\mu})\|\Delta\boldsymbol{\mu}\|_2, 1\right), \quad \forall v_i \in \mathbb{C}_- \end{cases}$$

where $\text{cos\_sim}(\boldsymbol{\mu}, \Delta\boldsymbol{\mu}) = \frac{\boldsymbol{\mu}^\top \Delta\boldsymbol{\mu}}{\|\boldsymbol{\mu}\|_2 \cdot \|\Delta\boldsymbol{\mu}\|_2}$. On the target graph, the expected accuracy is

$$\text{Acc}_T = \frac{1}{2}\Phi\left(\sqrt{\frac{d}{d+\gamma^2}} \cdot |1 + \gamma(2h-1)| \cdot \|\boldsymbol{\mu}\|_2 + \sqrt{\frac{d}{d+\gamma^2}} \cdot |1+\gamma| \cdot \text{cos\_sim}(\boldsymbol{\mu}, \Delta\boldsymbol{\mu})\|\Delta\boldsymbol{\mu}\|_2\right) +$$

$$\frac{1}{2}\Phi\left(\sqrt{\frac{d}{d+\gamma^2}} \cdot |1 + \gamma(2h-1)| \cdot \|\boldsymbol{\mu}\|_2 - \sqrt{\frac{d}{d+\gamma^2}} \cdot |1+\gamma| \cdot \text{cos\_sim}(\boldsymbol{\mu}, \Delta\boldsymbol{\mu})\|\Delta\boldsymbol{\mu}\|_2\right)$$

where $\Phi$ is the CDF of standard normal distribution. In order to compare the accuracy with the one in Corollary 3.2, we use Taylor expansion with Lagrange remainder. Let $x_0 = \sqrt{\frac{d}{d+\gamma^2}} \cdot |1 + \gamma(2h-1)| \cdot \|\boldsymbol{\mu}\|_2$ and $\Delta x = x - x_0 = \sqrt{\frac{d}{d+\gamma^2}} \cdot |1+\gamma| \cdot \text{cos\_sim}(\boldsymbol{\mu}, \Delta\boldsymbol{\mu})\|\Delta\boldsymbol{\mu}\|_2$. The Taylor series of $\Phi(x)$ at $x = x_0$ is:

$$\Phi(x) = \Phi(x_0) + \varphi(x_0)\Delta x + \frac{\varphi'(x_0 + \lambda\Delta x)}{2}(\Delta x)^2, \quad \exists \lambda \in (0,1)$$

where $\varphi(x) = \Phi'(x) = \frac{1}{\sqrt{2\pi}}e^{-\frac{1}{2}x^2}$ is the PDF of standard normal distribution and $\varphi'(c) = \Phi''(x)$ is the derivative of $\varphi(x)$. Therefore, the accuracy gap is:

$$\text{Acc}_S - \text{Acc}_T = \Phi(x) - \frac{1}{2}\Phi(x + \Delta x) - \frac{1}{2}\Phi(x - \Delta x)$$

$$= -\frac{\varphi'(x_0 + \lambda_1\Delta x) + \varphi'(x_0 - \lambda_2\Delta x)}{4} \cdot (\Delta x)^2, \quad \exists \lambda_1, \lambda_2 \in (0,1)$$

We finally give lower and upper bound of $-\frac{\varphi'(x_0 + \lambda_1\Delta x) + \varphi'(x_0 - \lambda_2\Delta x)}{4}$. Given $\|\Delta\boldsymbol{\mu}\|_2 \leq \frac{|1+\gamma(2h-1)|}{|1+\gamma|}\|\boldsymbol{\mu}\|_2$, we have $0 \leq \Delta x \leq x_0$ and thus $0 < x_0 - \lambda_2\Delta x < x_0 + \lambda_1\Delta x < 2x_0$.

When $0 < x < \infty$, we have $-\frac{1}{\sqrt{2\pi e}} \leq \varphi'(x) < 0$. Therefore we can give an upper bound of the constant:

$$-\frac{\varphi'(x_0 + \lambda_1 \Delta x) + \varphi'(x_0 - \lambda_2 \Delta x)}{4} \leq \frac{1}{2\sqrt{2\pi e}}$$

and also a lower bound

$$-\frac{\varphi'(x_0 + \lambda_1 \Delta x) + \varphi'(x_0 - \lambda_2 \Delta x)}{4} \geq -\frac{\varphi'(x_0 + \lambda_1 \Delta x)}{4} \geq -\frac{\max\{\varphi'(x_0), \varphi'(2x_0)\}}{4} > 0$$

Therefore, we have

$$\mathrm{Acc}_S - \mathrm{Acc}_T = \Theta((\Delta x)^2) = \Theta(\|\Delta \boldsymbol{\mu}\|_2^2)$$

We finally derive representation degradation and classifier bias. On the target graph, the optimal classifier is,

$$\boldsymbol{w} = \mathrm{sign}(1 + \gamma(2h - 1)) \cdot \frac{\boldsymbol{\mu}}{\|\boldsymbol{\mu}\|_2}, \quad b = -\mathrm{sign}(1 + \gamma(2h - 1)) \cdot (1 + \gamma) \cdot \mathrm{cos\_sim}(\boldsymbol{\mu}, \Delta \boldsymbol{\mu})\|\Delta \boldsymbol{\mu}\|_2$$

In this case, the distribution of $\boldsymbol{z}_i^\top \boldsymbol{w} + b$ will be identical to Eq. (11), and the accuracy will be identical to Eq. (12). It indicates that the representation degradation is $\Delta_f = 0$, and $\Delta_g = (\mathrm{Acc}_S - \mathrm{Acc}_T) - \Delta_f = \Theta(\|\Delta \boldsymbol{\mu}\|_2^2)$. $\qquad\square$

A.5 PROOF OF PROPOSITION 3.4

**Proposition 3.4** (Impacts of structure shifts). *When training a single-layer GCN on a source graph of CSBM$(\boldsymbol{\mu}, -\boldsymbol{\mu}, d_S, h_S)$, while testing it on a target graph of CSBM$(\boldsymbol{\mu}, -\boldsymbol{\mu}, d_T, h_T)$, where $1 \leq d_T = d_S - \Delta d < d_S$ and $\frac{1}{2} < h_T = h_S - \Delta h < h_S$, if $\gamma > 0$, we have*

$$\Delta_f = \Theta(\Delta h + \Delta d), \qquad \Delta_g = 0, \tag{5}$$

*which implies that the performance gap under structure shifts mainly attributes to the representation degradation.*

*Proof.* Without loss of generality, we consider a case with $\gamma > 0$, $\frac{1}{2} < h_T < h_S \leq 1$ and $1 \leq d_T < d_S \leq N$. In this case, decreases in both homophily and degree will lead to decreases in accuracy. Notice that our proposition can also be easily generalized to heterophilic setting.

We can reuse the results in Corollary 3.2. Given $\frac{1}{2} < h_T < h_S$, we have $\text{sign}(1 + \gamma(2h_S - 1)) = \text{sign}(1 + \gamma(2h_T - 1))$, and thus the optimal classifier we derived in Eq. (10) remains optimal on the target graph. Therefore, we have $\Delta_g = 0$, which means that the accuracy gap solely comes from the representation degradation. To calculate the accuracy gap, we consider the accuracy score as a function of degree $d$ and homophily $h$,

$$F\left(\begin{bmatrix} d \\ h \end{bmatrix}\right) = \Phi\left(\sqrt{\frac{d}{d + \gamma^2}} \cdot |1 + \gamma(2h - 1)| \cdot \|\boldsymbol{\mu}\|_2\right)$$

Its first order derivative is

$$\frac{\partial F}{\partial d} = \frac{\gamma^2}{2 d^{\frac{1}{2}} (d + \gamma^2)^{\frac{3}{2}}} \cdot |1 + \gamma(2h - 1)| \cdot \|\boldsymbol{\mu}\|_2 \cdot \varphi\left(\sqrt{\frac{d}{d + \gamma^2}} \cdot |1 + \gamma(2h - 1)| \cdot \|\boldsymbol{\mu}\|_2\right)$$

$$\frac{\partial F}{\partial h} = \sqrt{\frac{d}{d + \gamma^2}} \cdot 2\gamma \cdot \|\boldsymbol{\mu}\|_2 \cdot \varphi\left(\sqrt{\frac{d}{d + \gamma^2}} \cdot |1 + \gamma(2h - 1)| \cdot \|\boldsymbol{\mu}\|_2\right)$$

Both partial derivatives have lower and upper bounds, in the range of $h \in [\frac{1}{2}, 1], d \in [1, N]$:

$$\frac{\partial F}{\partial d} \leq \frac{\gamma^2}{2(1 + \gamma^2)^{\frac{3}{2}}} \cdot (1 + \gamma) \cdot \|\boldsymbol{\mu}\|_2 \cdot \frac{1}{\sqrt{2\pi}}$$

$$\frac{\partial F}{\partial d} \geq \frac{\gamma^2}{2 N^{\frac{1}{2}} (N + \gamma^2)^{\frac{3}{2}}} \cdot \|\boldsymbol{\mu}\|_2 \cdot \frac{1}{\sqrt{2\pi}}$$

$$\frac{\partial F}{\partial h} \leq 2\gamma \cdot \|\boldsymbol{\mu}\|_2 \cdot \frac{1}{\sqrt{2\pi}}$$

$$\frac{\partial F}{\partial h} \geq \sqrt{\frac{1}{1 + \gamma^2}} \cdot 2\gamma \cdot \|\boldsymbol{\mu}\|_2 \cdot \frac{1}{\sqrt{2\pi}}$$

Finally, to compare $\text{Acc}_S$ and $\text{Acc}_T$, we consider the Taylor expansion of $F$ at $\begin{bmatrix} d_S \\ h_S \end{bmatrix}$:

$$F\left(\begin{bmatrix} d_T \\ h_T \end{bmatrix}\right) = F\left(\begin{bmatrix} d_S - \Delta d \\ h_S - \Delta h \end{bmatrix}\right) = F\left(\begin{bmatrix} d_S \\ h_S \end{bmatrix}\right) - \nabla F\left(\begin{bmatrix} d_S - \lambda \Delta d \\ h_S - \lambda \Delta h \end{bmatrix}\right)^{\top} \begin{bmatrix} \Delta d \\ \Delta h \end{bmatrix}, \quad \exists \lambda \in (0, 1)$$

Therefore,

$$\text{Acc}_S - \text{Acc}_T = F\left(\begin{bmatrix} d_S \\ h_S \end{bmatrix}\right) - F\left(\begin{bmatrix} d_T \\ h_T \end{bmatrix}\right)$$

$$= \Theta(\Delta d + \Delta h)$$

and also $\Delta_f = (\text{Acc}_S - \text{Acc}_T) - \Delta_g = \Theta(\Delta d + \Delta h)$. $\qquad \square$

A.6 PROOF OF PROPOSITION 3.5

In this part, instead of treating $\gamma$ as fixed hyperparameter (as in Proposition 3.3 and 3.4), we now consider $\gamma$ as a trainable parameter that can be optimizer on both source and target graphs. We first derive the optimal $\gamma$ for a graph in Lemma A.2

**Lemma A.2.** *When training a single-layer GCN on a graph generated from CSBM($\boldsymbol{\mu}, -\boldsymbol{\mu}, d, h$), the optimal $\gamma$ that maximized the expected accuracy is $d(2h - 1)$.*

*Proof.* In Corollary 3.2, we have proved that with the optimal classifier, the accuracy is

$$\text{Acc} = \Phi \left( \sqrt{\frac{d}{d + \gamma^2}} \cdot |1 + \gamma(2h - 1)| \cdot \|\boldsymbol{\mu}\|_2 \right)$$

We then optimize $\gamma$ to reach the highest accuracy. Since $\Phi(x)$ is monotonely increasing, we only need to find the $\gamma$ that maximize $F(\gamma) = \frac{d}{d+\gamma^2}(1 + \gamma(2h - 1))^2$. Taking derivatives,

$$F'(\gamma) = \frac{d \cdot 2(1 + \gamma(2h - 1)) \cdot (2h - 1) \cdot (d + \gamma^2) - d(1 + \gamma(2h - 1))^2 \cdot 2\gamma}{(d + \gamma^2)^2}$$

$$= \frac{2d \cdot [1 + \gamma(2h - 1)] \cdot [(2h - 1)d - \gamma]}{(d + \gamma^2)^2}$$

$$\begin{cases} < 0, & \gamma \in (-\infty, -\frac{1}{2h-1}) \\ > 0, & \gamma \in (-\frac{1}{2h-1}, (2h - 1)d) \\ < 0, & \gamma \in ((2h - 1)d, +\infty) \end{cases}$$

Therefore, $F(\gamma)$ can only take maximal at $\gamma = (2h - 1)d$ or $\gamma \to -\infty$. We find that $\lim_{\gamma \to -\infty} F(\gamma) = (2h - 1)^2 d$ and $F((2h - 1)d) = 1 + (2h - 1)^2 d > (2h - 1)^2 d$. Therefore, the optimal $\gamma$ that maximize the accuracy is $\gamma = (2h - 1)d$, and the corresponding accuracy is

$$\text{Acc} = \Phi \left( \sqrt{1 + (2h - 1)^2 d} \cdot \|\boldsymbol{\mu}\|_2 \right)$$

$\square$

**Proposition 3.5** (Adapting $\gamma$). *Under the same learning setting as Proposition 3.4, adapting the source $\gamma_S$ to the optimal $\gamma_T = d_T(2h_T - 1)$ on the target graph can alleviate the representation degradation and improve the target classification accuracy by $\Theta((\Delta h)^2 + (\Delta d)^2)$.*

*Proof.* As shown in Lemma A.2, by adapting $\gamma$, the target accuracy can be improved from

$$\Phi \left( \sqrt{\frac{d_T}{d_T + \gamma_S^2}} \cdot |1 + \gamma_S(2h_T - 1)| \cdot \|\boldsymbol{\mu}\|_2 \right)$$

to

$$\Phi \left( \sqrt{\frac{d_T}{d_T + \gamma_T^2}} \cdot |1 + \gamma_T(2h_T - 1)| \cdot \|\boldsymbol{\mu}\|_2 \right) = \Phi \left( \sqrt{1 + (2h_T - 1)^2 d_T} \cdot \|\boldsymbol{\mu}\|_2 \right)$$

We know quantify this improvement. Let $F(\gamma) = \Phi \left( \sqrt{\frac{d_T}{d_T + \gamma^2}} \cdot |1 + \gamma(2h_T - 1)| \cdot \|\boldsymbol{\mu}\|_2 \right)$, since $\gamma_T$ is optimal on the target graph, we have $F'(\gamma_T) = 0$ and $F''(\gamma_T) < 0$. Therefore, we have

$$\Phi \left( \sqrt{1 + (2h_T - 1)^2 d_T} \cdot \|\boldsymbol{\mu}\|_2 \right) - \Phi \left( \sqrt{\frac{d_T}{d_T + \gamma_S^2}} \cdot |1 + \gamma_S(2h_T - 1)| \cdot \|\boldsymbol{\mu}\|_2 \right) = \Theta((\gamma_T - \gamma_S)^2)$$

Moreover, given $\gamma_S$ and $\gamma_T$ are optimal on source graph and target graph, respectively, we have $\gamma_S = 2(h_S - 1)d_S$ and $\gamma_T = 2(h_T - 1)d_T$, thereforem $|\gamma_T - \gamma_S| = \Theta(\Delta h + \Delta d)$. Therefore, the accuracy improvement is $\Theta((\Delta h)^2 + (\Delta d)^2)$. $\square$

A.7  GENERALIZATION TO MULTI-HOP GCNS

So far, our theoretical analysis has focused on single-layer GCNs and CSBM graphs, where ho-mophily and degree are used to parameterize structure shifts in one-hop neighborhoods. In this section, we extend our analysis to more general multi-hop scenarios.

For a graph $\mathcal{G}$ and a node $v_i$ on the graph, we define its $k$-hop neighbors as follows:

$$\mathbb{N}^{(k)}(v_i) = \begin{cases} \{v_i\}, & \text{if } k = 0, \\ \{v_j : \text{shortest\_distance}(v_j, v_i) = k\}, & \text{if } k \geq 1. \end{cases} \tag{13}$$

We further define

$$d^{(k)} = \frac{1}{N} d_i^{(k)}, \qquad \text{where } d_i^{(k)} = \left| \mathbb{N}_i^{(k)} \right|, \tag{14}$$

$$h^{(k)} = \frac{1}{N} h_i^{(k)}, \qquad \text{where } h_i^{(k)} = \frac{|\{v_j \in \mathbb{N}_i^{(k)} : y_j = y_i\}|}{d_i^{(k)}}. \tag{15}$$

For $k = 1$, $\mathbb{N}_i^{(1)}$ corresponds to the standard definition of node neighbors, and $d_i^{(1)}$ and $h_i^{(1)}$ align with node degree and homophily as defined in the main text. For $k \geq 2$, $d_i^{(k)}$ and $h_i^{(k)}$ capture more complex structure shifts in the egograph of $v_i$. For example:

- If the 2-hop egograph of $v_i$ only contains $d + 1$ nodes and all nodes are fully-connected (clustering coefficient is 1), then $d_i^{(2)} = 0$.
- If the 2-hop egograph of $v_i$ is a tree (clustering coefficient is 0) and each 1-hop neighbor of $v_i$ has degree $d$, then $d_i^{(2)} = d_i^{(1)} \cdot (d - 1)$.

This formulation allows us to model higher-order graph metrics, such as cluster coefficients, en-abling the analysis of more intricate structure shifts.

Inspired by methods like MixHop (Abu-El-Haija et al., 2019) and GPRGNN (Chien et al., 2021), we consider a $K$-hop GCN with a featurizer defined as:

$$\boldsymbol{z}_i = \sum_{k=1}^{K} \gamma_k \cdot \frac{1}{d_i^{(k)}} \sum_{v_j \in \mathbb{N}^{(k)}(v_i)} \boldsymbol{x}_j, \tag{16}$$

and the linear classifier is still $\boldsymbol{z}_i^\top \boldsymbol{w} + b$.

We denote $\boldsymbol{\gamma} = [\gamma_0, \cdots, \gamma_K]^\top$, and similarly,

$$\boldsymbol{d} = [d^{(0)}, \cdots, d^{(K)}]^\top, \qquad\qquad \boldsymbol{d}_i = [d_i^{(0)}, \cdots, d_i^{(K)}]^\top, \tag{17}$$

$$\boldsymbol{h} = [h^{(0)}, \cdots, h^{(K)}]^\top, \qquad\qquad \boldsymbol{h}_i = [h_i^{(0)}, \cdots, h_i^{(K)}]^\top. \tag{18}$$

**Proposition A.3.** *For graphs with two classes $\mathbb{C}_+$ and $\mathbb{C}_-$, and node attributes $\boldsymbol{x}_i \sim \mathcal{N}(\boldsymbol{\mu}_i, \boldsymbol{I})$ for each node $v_i$, where $\boldsymbol{\mu}_i = \boldsymbol{\mu}_+$ for $v_i \in \mathbb{C}_+$ and $\boldsymbol{\mu}_i = \boldsymbol{\mu}_-$ for $v_i \in \mathbb{C}_-$, the node representation $\boldsymbol{z}_i$ of node $v_i \in \mathbb{C}_+$ generated by a $K$-hop GCN follows a Gaussian distribution of*

$$\boldsymbol{z}_i \sim \mathcal{N}\left(\left(\boldsymbol{\gamma}^\top \boldsymbol{h}_i\right) \cdot \boldsymbol{\mu}_+ + \left(\boldsymbol{\gamma}^\top (\boldsymbol{1} - \boldsymbol{h}_i)\right) \cdot \boldsymbol{\mu}_-, \left(\boldsymbol{\gamma}^\top \operatorname{diag}(\boldsymbol{d}_i)^{-1} \boldsymbol{\gamma}\right) \cdot \boldsymbol{I}\right). \tag{19}$$

*When $\boldsymbol{\mu}_+ = \boldsymbol{\mu}$, $\boldsymbol{\mu}_- = -\boldsymbol{\mu}$, and all nodes have the same $\boldsymbol{d}_i$ and $\boldsymbol{h}_i$, the classifier maximizes the expected accuracy when $\boldsymbol{w} = \operatorname{sign}(\boldsymbol{\gamma}^\top(2\boldsymbol{h} - \boldsymbol{1})) \cdot \frac{\boldsymbol{\mu}}{\|\boldsymbol{\mu}\|_2}$ and $b = 0$. It gives a linear decision boundary of $\{\boldsymbol{z} : \boldsymbol{z}^\top \boldsymbol{w} = 0\}$ and the expected accuracy*

$$Acc = \Phi\left(\sqrt{\frac{(\boldsymbol{\gamma}^\top (2\boldsymbol{h} - \boldsymbol{1}))^2}{\boldsymbol{\gamma}^\top \operatorname{diag}(\boldsymbol{d}_i)^{-1} \boldsymbol{\gamma}}} \cdot \|\boldsymbol{\mu}\|_2\right), \tag{20}$$

*where $\Phi$ is the CDF of the standard normal distribution.*

Proposition A.3 is a natural extension of Proposition 3.1 and corollary 3.2 from one-layer GCN to multi-hop GCN. Based on this proposition, we can similarly derive conclusions analogous to Propositions 3.3 and 3.4 in the main text.

Finally, we show that the optimal $\boldsymbol{\gamma}$ should still be adapted based on $\boldsymbol{d}$ and $\boldsymbol{h}$, reinforcing the theo-retical insights provided earlier.

**Lemma A.4.** *Under the same setting as Proposition A.3, the optimal $\boldsymbol{\gamma}$ that maximized the expected accuracy is given by*

$$\boldsymbol{\gamma} \propto \mathrm{diag}(\boldsymbol{d})(2\boldsymbol{h} - \mathbf{1}) = \boldsymbol{d} \odot (2\boldsymbol{h} - \mathbf{1}), \tag{21}$$

*where $\odot$ is the element-wise multiplication. When $\gamma_0 = 1$, this yields:*

$$\gamma_k = d^{(k)}(2h^{(k)} - 1), \quad \forall k = 1, \cdots, K, \tag{22}$$

*which directly recovers the conclusion of Proposition 3.5.*

*Proof.* We aim to find $\boldsymbol{\gamma}$ that maximize the accuracy, equivalently, we optimize the following objective,

$$\boldsymbol{\gamma}^* = \arg\max_{\boldsymbol{\gamma}} \frac{\boldsymbol{\gamma}^\top (2\boldsymbol{h} - \mathbf{1})(2\boldsymbol{h} - \mathbf{1})^\top \boldsymbol{\gamma}}{\boldsymbol{\gamma}^\top \mathrm{diag}(\boldsymbol{d})^{-1} \boldsymbol{\gamma}}. \tag{23}$$

Let $\boldsymbol{\theta} = \mathrm{diag}(\boldsymbol{d})^{-\frac{1}{2}} \boldsymbol{\gamma}$, we solve

$$\boldsymbol{\theta}^* = \arg\max_{\boldsymbol{\theta}} \frac{\boldsymbol{\theta}^\top \mathrm{diag}(\boldsymbol{d})^{\frac{1}{2}} (2\boldsymbol{h} - \mathbf{1})(2\boldsymbol{h} - \mathbf{1})^\top \mathrm{diag}(\boldsymbol{d})^{\frac{1}{2}} \boldsymbol{\theta}}{\boldsymbol{\theta}^\top \boldsymbol{\theta}}. \tag{24}$$

Therefore, $\boldsymbol{\theta}^*$ is the first eigenvector of $\mathrm{diag}(\boldsymbol{d})^{\frac{1}{2}} (2\boldsymbol{h} - \mathbf{1})(2\boldsymbol{h} - \mathbf{1})^\top \mathrm{diag}(\boldsymbol{d})^{\frac{1}{2}}$, which is $\boldsymbol{\theta}^* \propto \mathrm{diag}(\boldsymbol{d})^{\frac{1}{2}} (2\boldsymbol{h} - \mathbf{1})$, and therefore $\boldsymbol{\gamma}^* \propto \mathrm{diag}(\boldsymbol{d})(2\boldsymbol{h} - \mathbf{1}) = \boldsymbol{d} \odot (2\boldsymbol{h} - \mathbf{1})$.

When $\gamma_0^* = 1$, since $d^{(0)} = 1$, $h^{(0)} = 1$, we have $\boldsymbol{\gamma}^* = \boldsymbol{d} \odot (2\boldsymbol{h} - \mathbf{1})$. $\qquad\square$

## A.8 PIC LOSS DECOMPOSITION

Notice that $\boldsymbol{\mu}_c = \frac{\sum_{i=1}^{N} \hat{\boldsymbol{Y}}_{i,c} \boldsymbol{z}_i}{\sum_{i=1}^{N} \hat{\boldsymbol{Y}}_{i,c}}, \forall c = 1, \cdots, C.$

$$
\begin{aligned}
\sigma^2 &= \sum_{i=1}^{N} \|\boldsymbol{z}_i - \boldsymbol{\mu}_*\|_2^2 \\
&= \sum_{i=1}^{N} \sum_{c=1}^{C} \hat{\boldsymbol{Y}}_{i,c} \|\boldsymbol{z}_i - \boldsymbol{\mu}_*\|_2^2 \\
&= \sum_{i=1}^{N} \sum_{c=1}^{C} \hat{\boldsymbol{Y}}_{i,c} \|\boldsymbol{z}_i - \boldsymbol{\mu}_c + \boldsymbol{\mu}_c - \boldsymbol{\mu}_*\|_2^2 \\
&= \sum_{i=1}^{N} \sum_{c=1}^{C} \hat{\boldsymbol{Y}}_{i,c} \|\boldsymbol{z}_i - \boldsymbol{\mu}_c\|_2^2 + 2 \sum_{i=1}^{N} \sum_{c=1}^{C} \hat{\boldsymbol{Y}}_{i,c} (\boldsymbol{z}_i - \boldsymbol{\mu}_c)^\top (\boldsymbol{\mu}_c - \boldsymbol{\mu}_*) + \sum_{i=1}^{N} \sum_{c=1}^{C} \hat{\boldsymbol{Y}}_{i,c} \|\boldsymbol{\mu}_c - \boldsymbol{\mu}_*\|_2^2 \\
&= \sum_{i=1}^{N} \sum_{c=1}^{C} \hat{\boldsymbol{Y}}_{i,c} \|\boldsymbol{z}_i - \boldsymbol{\mu}_c\|_2^2 + \sum_{i=1}^{N} \sum_{c=1}^{C} \hat{\boldsymbol{Y}}_{i,c} \|\boldsymbol{\mu}_c - \boldsymbol{\mu}_*\|_2^2 \\
&= \sigma_{\text{intra}}^2 + \sigma_{\text{inter}}^2
\end{aligned}
$$

## B CONVERGENCE ANALYSIS

### B.1 CONVERGENCE OF MATCHA

In this section, we give a convergence analysis of our Matcha framework. For the clarity of theoretical derivation, we first introduce the notation used in our proof.

- $z = \text{vec}(Z) \in \mathbb{R}^{ND}$ is the vectorization of node representations, where $Z \in \mathbb{R}^{N \times D}$ is the original node representation matrix, $N$ is the number of nodes, and $D$ is the dimensionality of representations.

- $\hat{y} = \text{vec}(\hat{Y}) \in \mathbb{R}^{NC}$ is the vectorization of predictions, where $\hat{Y} \in \mathbb{R}^{N \times C}$ is the original prediction of baseline TTA algorithm, given input $Z$, and $C$ is the number of classes.

- $h = \text{vec}(H) \in \mathbb{R}^{ND}$ is the vectorization of $H$, where $H = \text{MLP}(X) \in \mathbb{R}^{N \times D}$ is the (0-hop) node representations *before* propagation.

- $M = [\text{vec}(H), \text{vec}(\tilde{A}H), \cdots, \text{vec}(\tilde{A}^K H)] \in \mathbb{R}^{ND \times (K+1)}$ is the stack of 0-hop, 1-hop to $K$-hop representations.

- $\gamma \in \mathbb{R}^{K+1}$ is the hop-aggregation parameters for 0-hop, 1-hop to $K$-hop representations. Notice that $M\gamma = z$.

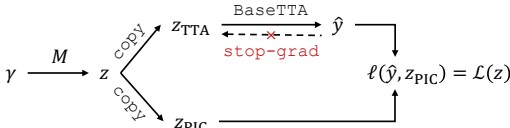

Figure 9: Computation graph of Matcha

Figure 9 gives a computation graph of Matcha.

- In the forward propagation, the node representation $z$ is copied into two copies, one ($z_{\text{TTA}}$) is used as the input of BaseTTA to obtain predictions $\hat{y}$, and the other ($z_{\text{PIC}}$) is used to calculate the PIC loss.

- In the backward propagation, since some baseline TTA algorithms do not support the evaluation of gradient, we do not compute the gradient through $z_{\text{TTA}}$, and only compute the gradient through $z_{\text{feat}}$. This introduces small estimation errors in the gradient, and thus introduces the challenge of convergence.

- We use

$$\nabla_z \mathcal{L}(z) = \frac{\partial \mathcal{L}(z)}{\partial z} = \frac{\partial \hat{y}}{\partial z_{\text{TTA}}} \frac{\partial \ell(\hat{y}, z_{\text{PIC}})}{\partial \hat{y}} + \frac{\partial \ell(\hat{y}, z_{\text{PIC}})}{\partial z_{\text{PIC}}}$$

to represent the "true" gradient of that consider the effects of both $z_{\text{TTA}}$ and $z_{\text{PIC}}$.

- Meanwhile, we use

$$\nabla_{z_{\text{PIC}}} \ell(\hat{y}, z_{\text{PIC}}) = \frac{\partial \ell(\hat{y}, z_{\text{PIC}})}{\partial z_{\text{PIC}}}$$

to represent the update direction of Matcha.

Clearly, the convergence of Matcha depends on the property of the baseline TTA algorithm BaseTTA. In the worst scenario, when the BaseTTA is unreliable and makes completely different predictions in each epoch, the convergence of Matcha could be challenging. However, in the more general case with mild assumptions on the loss function and baseline TTA algorithm, we show that Matcha can guarantee to converge. We start our proof by introducing assumptions.

**Assumption B.1** (Lipschitz and differentiable baseline TTA algorithm). The baseline TTA algorithm BaseTTA : $\mathbb{R}^{ND} \to \mathbb{R}^{ND}$ is differentiable and $L_1$-Lipschitz on $\mathcal{Z}$, i.e., there exists a constant $L_1$, s.t., for any $z_1, z_2 \in \mathcal{Z}$, where $\mathcal{Z} \subset \mathbb{R}^{ND}$ is the range of node representations,

$$\|\text{BaseTTA}(z_1) - \text{BaseTTA}(z_2)\|_2 \le L_1 \cdot \|z_1 - z_2\|_2$$

**Assumption B.2** (Lipschitz and differentiable loss function). The loss function $\ell(\hat{\boldsymbol{y}}, \boldsymbol{z}_{\text{PIC}}) : \mathbb{R}^{ND} \times \mathbb{R}^{ND} \to \mathbb{R}$ is differentiable and $L_2$-Lipschitz on $\mathcal{Y}$, i.e., there exists a constant $L_2$, s.t., for any $\hat{\boldsymbol{y}}_1, \hat{\boldsymbol{y}}_2 \in \mathcal{Y}$, where $\mathcal{Y} \subset \mathbb{R}^{ND}$ is the range of node predictions,

$$\|\ell(\hat{\boldsymbol{y}}_1, \boldsymbol{z}_{\text{PIC}}) - \ell(\hat{\boldsymbol{y}}_2, \boldsymbol{z}_{\text{PIC}})\|_2 \leq L_2 \cdot \|\hat{\boldsymbol{y}}_1 - \hat{\boldsymbol{y}}_2\|_2$$

*Remark* B.3. Assumption B.1 indicates that small changes in the input of TTA algorithm will not cause large change in its output, while Assumption B.2 indicates that small changes in the prediction will not significantly change the loss. These assumptions describe the robustness of the TTA algorithm and loss function. We verify in Lemma B.9 that standard linear layer followed by softmax activation satisfies these assumption.

**Definition B.4** ($\beta$-smoothness). A function $f : \mathbb{R}^d \to \mathbb{R}$ is $\beta$-smooth if for all $\boldsymbol{x}, \boldsymbol{y} \in \mathbb{R}^d$,

$$\|\nabla f(\boldsymbol{x}) - \nabla f(\boldsymbol{y})\|_2 \leq \beta \|\boldsymbol{x} - \boldsymbol{y}\|_2$$

equivalently, for all $\boldsymbol{x}, \boldsymbol{y} \in \mathbb{R}^d$,

$$f(\boldsymbol{y}) \leq f(\boldsymbol{x}) + \nabla f(\boldsymbol{x})^\top (\boldsymbol{y} - \boldsymbol{x}) + \frac{\beta}{2} \|\boldsymbol{x} - \boldsymbol{y}\|_2^2$$

**Assumption B.5** (Smooth loss function). The loss function $\mathcal{L}(\boldsymbol{z}) : \mathbb{R}^{ND} \to \mathbb{R}$ is $\beta$-smooth to $\boldsymbol{z}$.

*Remark* B.6. Assumption B.5 is a common assumption in the analysis of convergence (Bubeck, 2015).

**Lemma B.7** (Convergence of noisy SGD on smooth loss). *For any non-negative $L$-smooth loss function $\mathcal{L}(\boldsymbol{w})$ with parameters $\boldsymbol{w}$, conducting SGD with noisy gradient $\hat{g}(\boldsymbol{w})$ and step size $\eta = \frac{1}{L}$. If the gradient estimation error $\|\hat{g}(\boldsymbol{w}) - \nabla \mathcal{L}(\boldsymbol{w})\|_2^2 \leq \Delta^2$ for all $\boldsymbol{w}$, then for any weight initialization $\boldsymbol{w}^{(0)}$, after $T$ steps,*

$$\frac{1}{T} \sum_{t=0}^{T-1} \left\| \nabla \mathcal{L}(\boldsymbol{w}^{(t)}) \right\|_2^2 \leq 2\frac{L}{T} \mathcal{L}(\boldsymbol{w}^{(0)}) + \Delta^2$$

*Proof.* For any $\boldsymbol{w}^{(t)}$,

$$
\begin{aligned}
\mathcal{L}(\boldsymbol{w}^{(t+1)}) &\leq \mathcal{L}(\boldsymbol{w}^{(t)}) + \nabla \mathcal{L}(\boldsymbol{w}^{(t)})^\top (\boldsymbol{w}^{(t+1)} - \boldsymbol{w}^{(t)}) + \frac{L}{2} \left\| \boldsymbol{w}^{(t+1)} - \boldsymbol{w}^{(t)} \right\|_2^2 \quad \text{($L$-smoothness)} \\
&= \mathcal{L}(\boldsymbol{w}^{(t)}) + \nabla \mathcal{L}(\boldsymbol{w}^{(t)})^\top \left[ -\eta \left( \hat{g}(\boldsymbol{w}^{(t)}) - \nabla \mathcal{L}(\boldsymbol{w}^{(t)}) + \nabla \mathcal{L}(\boldsymbol{w}^{(t)}) \right) \right] \\
&\quad + \frac{L}{2} \left\| -\eta \left( \hat{g}(\boldsymbol{w}^{(t)}) - \nabla \mathcal{L}(\boldsymbol{w}^{(t)}) + \nabla \mathcal{L}(\boldsymbol{w}^{(t)}) \right) \right\|_2^2 \\
&= \mathcal{L}(\boldsymbol{w}^{(t)}) + \left( \frac{L\eta^2}{2} - \eta \right) \left\| \nabla \mathcal{L}(\boldsymbol{w}^{(t)}) \right\|_2^2 + \left( L\eta^2 - \eta \right) \nabla \mathcal{L}(\boldsymbol{w}^{(t)})^\top \left( \hat{g}(\boldsymbol{w}^{(t)}) - \nabla \mathcal{L}(\boldsymbol{w}^{(t)}) \right) \\
&\quad + \frac{L\eta^2}{2} \left\| \hat{g}(\boldsymbol{w}^{(t)}) - \nabla \mathcal{L}(\boldsymbol{w}^{(t)}) \right\|_2^2 \\
&= \mathcal{L}(\boldsymbol{w}^{(t)}) - \frac{1}{2L} \left\| \nabla \mathcal{L}(\boldsymbol{w}^{(t)}) \right\|_2^2 + \frac{1}{2L} \left\| \hat{g}(\boldsymbol{w}^{(t)}) - \nabla \mathcal{L}(\boldsymbol{w}^{(t)}) \right\|_2^2 \quad \text{($\eta = \frac{1}{L}$)}
\end{aligned}
$$

Equivalently,

$$\left\| \nabla \mathcal{L}(\boldsymbol{w}^{(t)}) \right\|_2^2 \leq 2L \left( \mathcal{L}(\boldsymbol{w}^{(t)}) - \mathcal{L}(\boldsymbol{w}^{(t+1)}) \right) + \left\| \hat{g}(\boldsymbol{w}^{(t)}) - \nabla \mathcal{L}(\boldsymbol{w}^{(t)}) \right\|_2^2$$

Average over $t = 0, \cdots, T-1$, we get

$$
\begin{aligned}
\frac{1}{T} \sum_{t=0}^{T-1} \left\| \nabla \mathcal{L}(\boldsymbol{w}^{(t)}) \right\|_2^2 &\leq 2\frac{L}{T} \left( \mathcal{L}(\boldsymbol{w}^{(0)}) - \mathcal{L}(\boldsymbol{w}^{(T)}) \right) + \frac{1}{T} \sum_{t=0}^{T-1} \left\| \hat{g}(\boldsymbol{w}^{(t)}) - \nabla \mathcal{L}(\boldsymbol{w}^{(t)}) \right\|_2^2 \\
&\leq 2\frac{L}{T} \mathcal{L}(\boldsymbol{w}^{(0)}) + \frac{1}{T} \sum_{t=0}^{T-1} \left\| \hat{g}(\boldsymbol{w}^{(t)}) - \nabla \mathcal{L}(\boldsymbol{w}^{(t)}) \right\|_2^2 \\
&\leq 2\frac{L}{T} \mathcal{L}(\boldsymbol{w}^{(0)}) + \Delta^2
\end{aligned}
$$

$\square$

Lemma B.7 gives a general convergence guarantee of noisy gradient descent on smooth functions. Next, in Theorem B.8, we give the convergence analysis of `Matcha`.

**Theorem B.8** (Convergence of `Matcha`). *With Assumption B.1, B.2 and B.5 held, if we start with $\gamma^{(0)}$ and conduct $T$ steps of gradient descent with $\nabla_{z_{PIC}}\ell(\hat{y}, z_{PIC})$, and step size $\frac{1}{\beta\|M\|_2^2}$, we have*

$$\frac{1}{T}\sum_{t=0}^{T-1}\left\|\nabla_\gamma\mathcal{L}(\gamma^{(t)})\right\|_2^2 \le 2\frac{\beta\|M\|_2^2}{T}\mathcal{L}(\gamma^{(0)}) + L_1^2 L_2^2 \|M\|_2^2$$

*Proof.* We first give an upper bound of the gradient estimation error

$$
\begin{aligned}
\|\nabla_{z_{\mathrm{PIC}}}\ell(\hat{y}, z_{\mathrm{PIC}}) - \nabla_z\ell(\hat{y}, z_{\mathrm{PIC}})\|_2 &= \|\nabla_{z_{\mathrm{TTA}}}\ell(\hat{y}, z_{\mathrm{PIC}})\|_2 \\
&= \left\|\frac{\partial\hat{y}}{\partial z_{\mathrm{TTA}}}\frac{\partial\ell(\hat{y}, z_{\mathrm{PIC}})}{\partial\hat{y}}\right\|_2 \\
&\le \left\|\frac{\partial\hat{y}}{\partial z_{\mathrm{TTA}}}\right\|_2 \cdot \left\|\frac{\partial\ell(\hat{y}, z_{\mathrm{PIC}})}{\partial\hat{y}}\right\|_2 \\
&\le L_1 \cdot L_2 \qquad\qquad \text{(Assumption B.1, B.2)}
\end{aligned}
$$

Therefore, the gradient estimation error can be bounded by $L_1 \cdot L_2 \cdot \|M\|_2$.

Meanwhile, since the loss function is $\beta$-smooth w.r.t. $z$, it is $\beta \cdot \|M\|_2^2$-smooth to $\gamma$ since

$$
\begin{aligned}
\|\nabla_\gamma\mathcal{L}(\gamma_1) - \nabla_\gamma\mathcal{L}(\gamma_2)\|_2 &= \|M^\top(\nabla_{z_1}\mathcal{L}(\gamma_1) - \nabla_{z_2}\mathcal{L}(\gamma_2))\|_2 \\
&\le \|M\|_2 \cdot \beta \cdot \|z_1 - z_2\|_2 \\
&\le \|M\|_2^2 \cdot \beta \cdot \|\gamma_1 - \gamma_2\|_2
\end{aligned}
$$

Finally, by Lemma B.7, we have

$$\frac{1}{T}\sum_{t=0}^{T-1}\left\|\nabla_\gamma\mathcal{L}(\gamma^{(t)})\right\|_2^2 \le 2\frac{\beta\|M\|_2^2}{T}\mathcal{L}(\gamma^{(0)}) + L_1^2 L_2^2 \|M\|_2^2$$

$\square$

## B.2 EXAMPLE: LINEAR LAYER FOLLOWED BY SOFTMAX

**Lemma B.9.** *When using a linear layer followed by a softmax as the* BaseTTA*, the function* $\ell(\hat{y}, z_{PIC})$*, as a function of* $z_{TTA}$*, is* $(2\|W\|_2)$*-Lipschitz, where* $W$ *is the weights for the linear layer.*

*Proof.* We manually derive the gradient of $\ell(\hat{y}, z_{\text{PIC}})$ w.r.t. $z_{\text{TTA}}$. Denote $a \in \mathbb{R}^{NC}$ as the output of the linear layer, $a_i$ as the linear layer output for the $i$-th node, and $a_{ic}$ as its $c$-th element (corresponding to label $c$). We have:

$$
\begin{aligned}
\frac{\partial \ell}{\partial a_{ic}} &= \sum_{c'=1}^{C} \frac{\partial \hat{Y}_{i,c'}}{\partial a_{ic}} \cdot \frac{\partial \ell}{\partial \hat{Y}_{i,c'}} \\
&= (\hat{Y}_{i,c} - \hat{Y}_{i,c}^2) \frac{\|z_i - \mu_c\|_2^2}{\sum_{i=1}^{N} \|z_i - \mu_*\|_2^2} + \sum_{c' \neq c} (-\hat{Y}_{i,c} \hat{Y}_{i,c'}) \frac{\|z_i - \mu_{c'}\|_2^2}{\sum_{i=1}^{N} \|z_i - \mu_*\|_2^2} \\
&= \frac{\hat{Y}_{i,c} \|z_i - \mu_c\|_2^2}{\sum_{i=1}^{N} \|z_i - \mu_*\|_2^2} - \hat{Y}_{i,c} \frac{\sum_{c'=1}^{C} \hat{Y}_{i,c'} \|z_i - \mu_{c'}\|_2^2}{\sum_{i=1}^{N} \|z_i - \mu_*\|_2^2}
\end{aligned}
$$

Therefore, as vector representation:

$$
\begin{aligned}
\left\| \frac{\partial \ell}{\partial a} \right\|_2 &\leq \left\| \left[ \frac{\hat{Y}_{i,c} \|z_i - \mu_c\|_2^2}{\sum_{i=1}^{N} \|z_i - \mu_*\|_2^2} \right]_{i,c} \right\|_2 + \left\| \left[ \hat{Y}_{i,c} \frac{\sum_{c'=1}^{C} \hat{Y}_{i,c'} \|z_i - \mu_{c'}\|_2^2}{\sum_{i=1}^{N} \|z_i - \mu_*\|_2^2} \right]_{i,c} \right\|_2 \\
&\leq \left\| \left[ \frac{\hat{Y}_{i,c} \|z_i - \mu_c\|_2^2}{\sum_{i=1}^{N} \|z_i - \mu_*\|_2^2} \right]_{i,c} \right\|_1 + \left\| \left[ \hat{Y}_{i,c} \frac{\sum_{c'=1}^{C} \hat{Y}_{i,c'} \|z_i - \mu_{c'}\|_2^2}{\sum_{i=1}^{N} \|z_i - \mu_*\|_2^2} \right]_{i,c} \right\|_1 \\
&= \frac{\sum_{i=1}^{N} \sum_{c=1}^{C} \hat{Y}_{i,c} \|z_i - \mu_c\|_2^2}{\sum_{i=1}^{N} \|z_i - \mu_*\|_2^2} + \sum_{i=1}^{N} \sum_{c=1}^{C} \hat{Y}_{i,c} \frac{\sum_{c'=1}^{C} \hat{Y}_{i,c'} \|z_i - \mu_{c'}\|_2^2}{\sum_{i=1}^{N} \|z_i - \mu_*\|_2^2} \\
&= \frac{\sum_{i=1}^{N} \sum_{c=1}^{C} \hat{Y}_{i,c} \|z_i - \mu_c\|_2^2}{\sum_{i=1}^{N} \|z_i - \mu_*\|_2^2} + \frac{\sum_{i=1}^{N} \sum_{c'=1}^{C} \hat{Y}_{i,c'} \|z_i - \mu_{c'}\|_2^2}{\sum_{i=1}^{N} \|z_i - \mu_*\|_2^2} \\
&= 2 \cdot \frac{\sigma_{\text{intra}}^2}{\sigma^2} \\
&\leq 2
\end{aligned}
$$

Notice that although the computation of $\mu_c$ also uses $\hat{Y}_{i,c}$,

$$
\frac{\partial \ell}{\partial \mu_c} = \frac{2}{\sigma^2} \sum_{i=1}^{N} \hat{Y}_{i,c} (\mu_c - z_i) = 0
$$

So there are no back propagating gradients through $\ell \to \mu_c \to \hat{Y}_{i,c}$.

Finally, because for each node $v_i$, $a_i = W^\top z_{\text{TTA},i}$, we have

$$
\left\| \frac{\partial \ell}{\partial z_{\text{TTA}}} \right\|_2 \leq \|W\|_2 \cdot \left\| \frac{\partial \ell}{\partial a} \right\|_2 = 2\|W\|_2
$$

$\square$

*Remark* B.10. Lemma B.9 verifies assumption B.1 and B.2: $L_1 \cdot L_2 = 2\|W\|_2$. It also reveals the benefit of using soft-predictions instead of hard-predictions. Hard predictions can be seen as scaling up $W$. In this case, the Lipschitz constant will be much larger or even unbounded, which impedes the convergence of Matcha.

## C    ADDITIONAL EXPERIMENTS

### C.1    EFFECT OF ATTRIBUTE SHIFTS AND STRUCTURE SHIFTS

We empirically verify that attribute shifts and structure shifts impact the GNN's accuracy on target graph in different ways. We use t-SNE to visualize the node representations on CSBM dataset under attribute shifts and structure shifts (homophily shifts). As shown in Figure 10, under attribute shift (c), although the node representations are shifted from the source graph, two classes are still mostly discriminative, which is similar to the case without distribution shifts (c). However, under homophily shift (d), the node representations for two classes mix together. These results match with our theoretical analysis in Propositions 3.3 and 3.4.

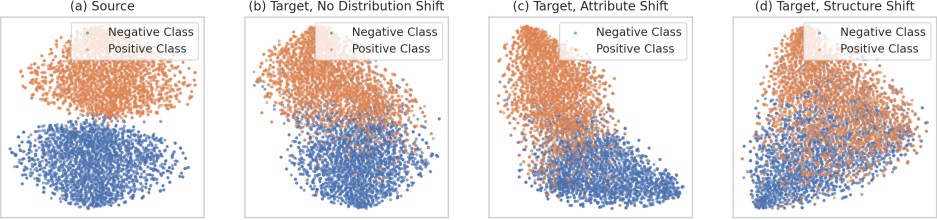

Figure 10: t-SNE visualization of node representations on CSBM dataset.

### C.2    ROBUSTNESS TO NOISY PREDICTION

In `Matcha`, representation quality and prediction accuracy mutually reinforce each other throughout the adaptation process. A natural question arises: if the model's predictions contain significant noise before adaptation, can `Matcha` still be effective? To address this, we conducted an empirical study on the CSBM dataset with severe homophily shift. We visualize the logits distribution for two classes of nodes in Figure 11.

- Before adaptation, the predictions exhibit significant noise, with substantial overlap in the logits of two classes.

- However, as adaptation progresses, `Matcha` is still able to gradually refine the node representations and improve accuracy.

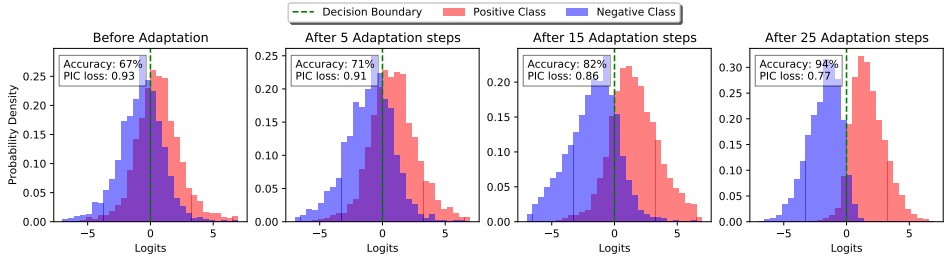

Figure 11: `Matcha` improves accuracy even when the initial predictions are highly noisy

## C.3 DIFFERENT LEVELS OF STRUCTURE SHIFT

In the main text, we evaluated the performance of `Matcha` under both homophily and degree shifts. In this section, we extend our evaluation by testing `Matcha` across varying degrees of these structure shifts. For each scenario (e.g., homophily: homo $\rightarrow$ hetero, hetero $\rightarrow$ homo, and degree: high $\rightarrow$ low, low $\rightarrow$ high), we manipulate either the homophily or degree of the source graph while keeping the target graph fixed, thereby creating different levels of homophily or degree shifts. The larger the discrepancy between the source and target graphs in terms of homophily or degree, the greater the level of structure shift. For instance, a shift from $0.6 \rightarrow 0.2$ indicates training a model on a source graph with homophily 0.6 and evaluating it on a target graph with homophily 0.2. By comparison, a shift from $0.8 \rightarrow 0.2$ represents a more substantial homophily shift.

The results of our experiments are summarized in Tables 3 and 4. Across all four settings, as the magnitude of the structure shift increases, the performance of GNNs trained using ERM declines significantly. However, under all settings, `Matcha` consistently improves model performance. For example, in the homo $\rightarrow$ hetero setting, when the homophily gap increases from 0.4 (0.6 - 0.2) to 0.6 (0.8 - 0.2), the accuracy of ERM-trained models decreases by over 16%, while the accuracy of models trained with `Matcha` declines by less than 2%. This demonstrates that `Matcha` effectively mitigates the negative impact of structure shifts on GNNs.

Table 3: Accuracy (mean $\pm$ s.d. %) on CSBM under different levels of *homophily shift*

| Method | homo $\rightarrow$ hetero | | | hetero $\rightarrow$ homo | | |
|---|---|---|---|---|---|---|
| | $0.4 \rightarrow 0.2$ | $0.6 \rightarrow 0.2$ | $0.8 \rightarrow 0.2$ | $0.2 \rightarrow 0.8$ | $0.4 \rightarrow 0.8$ | $0.6 \rightarrow 0.8$ |
| ERM | $90.05 \pm 0.15$ | $82.51 \pm 0.28$ | $73.62 \pm 0.44$ | $76.72 \pm 0.89$ | $83.55 \pm 0.50$ | $89.34 \pm 0.03$ |
| + Matcha | $90.79 \pm 0.17$ | $89.55 \pm 0.21$ | $89.71 \pm 0.27$ | $90.68 \pm 0.26$ | $90.59 \pm 0.24$ | $91.14 \pm 0.17$ |

Table 4: Accuracy (mean $\pm$ s.d. %) on CSBM under different levels of *degree shift*

| Method | high $\rightarrow$ low | | | low $\rightarrow$ high | | |
|---|---|---|---|---|---|---|
| | $5 \rightarrow 2$ | $10 \rightarrow 2$ | $20 \rightarrow 2$ | $2 \rightarrow 20$ | $5 \rightarrow 20$ | $10 \rightarrow 20$ |
| ERM | $88.67 \pm 0.13$ | $86.47 \pm 0.38$ | $85.55 \pm 0.12$ | $93.43 \pm 0.37$ | $95.35 \pm 0.84$ | $97.31 \pm 0.36$ |
| + Matcha | $88.78 \pm 0.13$ | $88.55 \pm 0.44$ | $88.10 \pm 0.21$ | $97.01 \pm 1.00$ | $97.24 \pm 1.11$ | $97.89 \pm 0.25$ |

## C.4 EVOLVING TARGET GRAPHS

Our experiments in the main text mainly focused on adapting to a single target graph with a stable distribution. In this section, we test `Matcha` on a stream of target graphs with evolving structure shifts. Specifically, we used the Syn-Cora dataset, where the model was pre-trained on a source graph with a homophily of 0.8. It was then sequentially adapted to five target graphs with homophilies of 0.1, 0.7, 0.3, 0.9, and 0.2, simulating continuously changing homophily. The experimental setup, except for the sequence of target graphs, was identical to that in the Syn-Cora experiments described in the main text.

For example, in the column corresponding to the target graph with a homophily of 0.3, we compared two scenarios: (1) static: directly adapting from $h = 0.8$ to $h = 0.3$, and (2) evolving: sequentially adapting through $(h = 0.8) \rightarrow (h = 0.1) \rightarrow (h = 0.7) \rightarrow (h = 0.3)$.

Table 5: Accuracy (mean $\pm$ s.d. %) on Syn-Cora with *static* target graph and *evolving* target graph

| Setting | Method | Target graph | | | | |
|---|---|---|---|---|---|---|
| | | $h = 0.1$ | $h = 0.7$ | $h = 0.3$ | $h = 0.9$ | $h = 0.2$ |
| Static | ERM | $63.19 \pm 1.28$ | $88.88 \pm 0.61$ | $71.46 \pm 0.62$ | $97.15 \pm 0.32$ | $65.67 \pm 0.35$ |
| Static | Matcha | $79.75 \pm 1.04$ | $90.57 \pm 0.47$ | $79.68 \pm 0.73$ | $97.40 \pm 0.28$ | $78.96 \pm 1.08$ |
| Evolving | Matcha | $79.75 \pm 1.04$ | $90.65 \pm 0.33$ | $77.43 \pm 0.62$ | $97.31 \pm 0.42$ | $78.26 \pm 1.02$ |

The results, shown in Table 5 above, indicate that `Matcha` achieves performance on evolving graphs highly comparable to that on static graphs. This demonstrates `Matcha`'s ability to handle dynamic scenarios effectively, even under continuously changing graph structures.

## C.5  ROBUSTNESS TO ADDITIONAL ADVERSARIAL SHIFT

While `Matcha` primarily targets natural structure shifts, inspired by (Jin et al., 2023), we test the robustness of `Matcha` against adversarial attacks by applying the PR-BCD attack (Geisler et al., 2021) on the target graph in our Syn-Cora experiments, varying the perturbation rate from 5% to 20%. The results are shown in Table 6. We found that while the accuracy of ERM dropped by 20.2%, the performance of `Matcha` only decreased by 2.3%. This suggests that our algorithm has some robustness to adversarial attacks, possibly due to the overlap between adversarial attacks and structure shifts. Specifically, we observed a decrease in homophily in the target graph under adversarial attack, indicating a similarity to structure shifts.

Table 6: Accuracy (%) on Syn-Cora with additional adversarial shift

| Perturbation rate | No attack | 5% | 10% | 15% | 20% |
|---|---|---|---|---|---|
| ERM | 65.67 | 60.00 | 55.25 | 50.22 | 45.47 |
| + Matcha | 78.96 | 78.43 | 78.17 | 77.21 | 76.61 |
| Homophily | 0.2052 | 0.1923 | 0.1800 | 0.1690 | 0.1658 |

## C.6  ABLATION STUDY WITH DIFFERENT LOSS FUNCTIONS

We compare our proposed PIC loss with two existing surrogate losses: entropy (Wang et al., 2021) and pseudo-label (Liang et al., 2020). While PIC loss use the ratio form of $\sigma^2_{\text{intra}}$ and $\sigma^2_{\text{inter}}$, we also compare it with a difference form $\sigma^2_{\text{intra}} - \sigma^2_{\text{inter}}$, which also encourage larger $\sigma^2_{\text{inter}}$ and smaller $\sigma^2_{\text{intra}}$. The results are shown in Table 7: Our PIC loss has better performance under four structure shift scenarios.

Table 7: Accuracy (mean $\pm$ s.d. %) on CSBM with different losses.

| Loss | Homophily shift | | Degree shift | |
|---|---|---|---|---|
| | homo $\rightarrow$ hetero | hetero $\rightarrow$ homo | high $\rightarrow$ low | low $\rightarrow$ high |
| (None) | 73.62 $\pm$ 0.44 | 76.72 $\pm$ 0.89 | 86.47 $\pm$ 0.38 | 92.92 $\pm$ 0.43 |
| Entropy | 75.89 $\pm$ 0.68 | 89.98 $\pm$ 0.23 | 86.81 $\pm$ 0.34 | 93.75 $\pm$ 0.72 |
| PseudoLabel | 77.29 $\pm$ 3.04 | 89.44 $\pm$ 0.22 | 86.72 $\pm$ 0.31 | 93.68 $\pm$ 0.69 |
| $\sigma^2_{\text{intra}} - \sigma^2_{\text{inter}}$ | 76.10 $\pm$ 0.43 | 72.43 $\pm$ 0.65 | 82.56 $\pm$ 0.99 | 92.92 $\pm$ 0.44 |
| PIC (Ours) | **89.71 $\pm$ 0.27** | **90.68 $\pm$ 0.26** | **88.55 $\pm$ 0.44** | **93.78 $\pm$ 0.74** |

## C.7  HYPERPARAMETER SENSITIVITY WITH DIFFERENT NUMBER OF GPR STEPS $K$

Although the `Matcha` does not involve any hyperparameters other than the learning rate $\eta$ and number of adaptation rounds $T$, it may be combined with GNN models with different dimension of $\gamma$. Therefore in this part, we combine `Matcha` with GPRGNN models using different $K$, i.e., the number of GPR steps, to test the robustness to different hyperparameter selection of the GNN model. Specifically, we tried values of $K$ ranging from 3 to 15 on Syn-Cora and Syn-Products datasets. Notice that in our experiments in 5.1, we use $K = 9$. As shown in Table 8, `Matcha` remains effective under a wide range of $K$.

Table 8: Hyperparameter sensitivity of $K$

| Dataset | Method | $K$ | | | | | | |
|---|---|---|---|---|---|---|---|---|
| | | 3 | 5 | 7 | 9 | 11 | 13 | 15 |
| Syn-Cora | ERM | 64.18 $\pm$ 0.72 | 65.69 $\pm$ 0.88 | 66.01 $\pm$ 0.89 | 65.67 $\pm$ 0.35 | 65.36 $\pm$ 0.66 | 64.47 $\pm$ 1.54 | 64.91 $\pm$ 0.97 |
| | + Matcha | 81.35 $\pm$ 0.64 | 80.13 $\pm$ 0.59 | 79.50 $\pm$ 0.72 | 78.96 $\pm$ 1.08 | 78.42 $\pm$ 0.85 | 78.60 $\pm$ 0.81 | 77.92 $\pm$ 0.87 |
| Syn-Products | ERM | 42.69 $\pm$ 1.03 | 41.86 $\pm$ 2.11 | 39.71 $\pm$ 2.75 | 37.52 $\pm$ 2.93 | 35.06 $\pm$ 2.27 | 33.17 $\pm$ 2.38 | 35.57 $\pm$ 0.55 |
| | + Matcha | 72.09 $\pm$ 0.50 | 71.42 $\pm$ 0.65 | 70.58 $\pm$ 1.01 | 69.69 $\pm$ 1.06 | 69.48 $\pm$ 1.16 | 69.35 $\pm$ 0.66 | 69.72 $\pm$ 0.70 |

## C.8 COMPUTATION TIME

Due to the need to adapt hop-aggregation parameters $\gamma$, `Matcha` inevitably introduces additional computation costs, which vary depending on the chosen model, target graph, and base TTA algorithm. We documented the computation times for each component of ERM + `Matcha` and T3A + `Matcha` in our CSBM experiments:

- *Initial inference* involves the time required for the model's first prediction on the target graph, including the computation of 0-hop to $K$-hop representations $\{\boldsymbol{H}^{(0)}, \cdots, \boldsymbol{H}^{(K)}\}$, their aggregation into $\boldsymbol{Z} = \sum_{k=0}^{K} \gamma_k \boldsymbol{H}^{(k)}$, and prediction using a linear layer classifier. **This is also the time required for a direct prediction without any adaptation.** $\{\boldsymbol{H}^{(0)}, \cdots, \boldsymbol{H}^{(K)}\}$ is cached in the initial inference.

- *Adaptation (for each epoch)* accounts for the time required for each step of adaptation after the initial inference, and includes four stages:

  - *Forward pass* involves calculation of $\boldsymbol{Z}$ using the current $\gamma$ and cached $\{\boldsymbol{H}^{(0)}, \cdots, \boldsymbol{H}^{(K)}\}$, and prediction using the linear layer classifier (or with T3A algorithm). Since `Matcha` only updates $\gamma$, $\{\boldsymbol{H}^{(0)}, \cdots, \boldsymbol{H}^{(K)}\}$ can be cached without recomputation in each epoch. Note that other TTA algorithms could also adopt the same or similar caching strategies.

  - *Computing PIC loss* involves calculating PIC loss using node representations $\boldsymbol{Z}$ and the predictions $\hat{\boldsymbol{Y}}$.

  - *Back propagation* computes the gradients with respect to $\gamma$. Similarly, as only $\gamma$ is updated, there is no need for full GNN back propagation.

  - *Updating parameters*, i.e., $\gamma$, with the computed gradients.

Table 9: Computation time on CSBM

| Method | Stage | Computation time (ms) | Additional computation time |
|---|---|---|---|
| - | Initial Inference | $27.687 \pm 0.413$ | - |
| GTrans | Adaptation (for each epoch) | $134.457 \pm 2.478$ | 485.63% |
| SOGA | Adaptation (for each epoch) | $68.500 \pm 13.354$ | 247.41% |
| ERM + `Matcha` | Adaptation (for each epoch) | $3.292 \pm 0.254$ | 11.89% |
| | - Forward pass | $1.224 \pm 0.131$ | 4.42% |
| | - Computing PIC loss | $0.765 \pm 0.019$ | 2.76% |
| | - Back-propagation | $1.189 \pm 0.131$ | 4.30% |
| | - Updating parameter | $0.113 \pm 0.001$ | 0.41% |
| T3A + `Matcha` | Adaptation (for each epoch) | $6.496 \pm 0.333$ | 23.46% |
| | - Forward pass | $4.464 \pm 0.248$ | 16.12% |
| | - Computing PIC loss | $0.743 \pm 0.011$ | 2.68% |
| | - Back-propagation | $1.174 \pm 0.167$ | 4.24% |
| | - Updating parameter | $0.115 \pm 0.004$ | 0.41% |

We provide the computation time for each stage in Table 9 above. While the initial inference time is 27.689 ms, each epoch of adaptation only introduce 3.292 ms (6.496 ms) additional computation time when combined with ERM (T3A), which is only 11.89% (23.46%) of the initial inference. This superior efficiency comes from (1) `Matcha` only updating the hop-aggregation parameters and (2) the linear complexity of our PIC loss.

We also compare the computation time of `Matcha` with other graph TTA algorithms. A significant disparity is observed: while the computation time for each step of adaptation in other graph TTA algorithms is several times that of inference, the adaptation time of our algorithm is merely 1/9 (1/4) of the inference time, making it almost negligible in comparison.

## C.9 SCALABILITY

In the end of Section 4, we show the computational complexity of `Matcha` is linear to the number of nodes. To further validate this, we have conducted experiments on graphs of varying sizes from 1 million to 10 million nodes, and record the computation time for each epoch of adaptation. The results confirm that the computation time for `Matcha` indeed scales linearly with graph size, demonstrating its efficiency even for very large graphs.

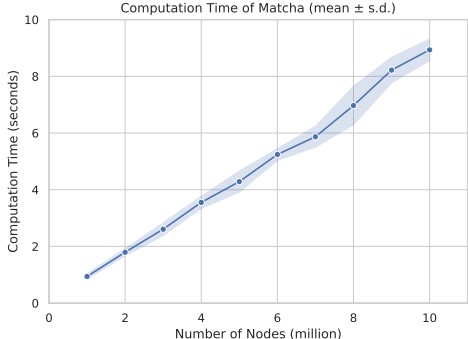

Figure 12: Scalability of `Matcha`

## C.10 MORE ARCHITECTURES

Besides GPRGNN (Chien et al., 2021), our proposed `Matcha` framework can also be integrated to more GNN architectures. We conduct experiments on Syn-cora dataset with three additional GNNs: APPNP (Klicpera et al., 2019), JKNet (Xu et al., 2018), and GCNII (Chen et al., 2020).

- For APPNP, we adapt the teleport probability $\alpha$.
- For JKNet, we use weighted average as the layer aggregation, and adapt the weights for each intermediate representations.
- For GCNII, we adapt the hyperparameter $\alpha_l$ for each layer.

Notice that Tent can only be applied to models with batch normalization layers, which are not included for JKNet in GCNII in our implementation.

Table 10: Accuracy (mean $\pm$ s.d.) on Syn-Cora with different GNN architectures

| Method | GPRGNN | APPNP | JKNet | GCNII |
|---|---|---|---|---|
| ERM | $65.67 \pm 0.35$ | $70.24 \pm 0.88$ | $47.87 \pm 0.90$ | $67.95 \pm 1.33$ |
| + `Matcha` | $78.96 \pm 1.08$ | $80.63 \pm 0.35$ | $51.57 \pm 2.09$ | $74.33 \pm 0.45$ |
| T3A | $68.25 \pm 1.10$ | $70.98 \pm 0.86$ | $47.93 \pm 0.85$ | $68.20 \pm 1.31$ |
| + `Matcha` | $78.40 \pm 1.04$ | $80.70 \pm 0.38$ | $51.84 \pm 1.87$ | $74.96 \pm 0.23$ |
| Tent | $66.26 \pm 0.38$ | $70.15 \pm 1.08$ | - | - |
| + `Matcha` | $78.87 \pm 1.07$ | $80.72 \pm 0.18$ | - | - |
| AdaNPC | $67.34 \pm 0.76$ | $70.53 \pm 0.76$ | $47.93 \pm 0.77$ | $68.39 \pm 1.18$ |
| + `Matcha` | $77.45 \pm 0.62$ | $80.11 \pm 0.61$ | $48.32 \pm 0.69$ | $74.44 \pm 0.35$ |
| GTrans | $68.60 \pm 0.32$ | $73.50 \pm 0.62$ | $51.38 \pm 0.58$ | $74.08 \pm 1.26$ |
| + `Matcha` | $83.49 \pm 0.78$ | $85.17 \pm 0.43$ | $53.76 \pm 2.26$ | $80.50 \pm 0.40$ |
| SOGA | $67.16 \pm 0.72$ | $78.62 \pm 0.48$ | $47.96 \pm 0.55$ | $66.87 \pm 1.50$ |
| + `Matcha` | $79.03 \pm 1.10$ | $80.88 \pm 0.56$ | $52.05 \pm 1.64$ | $74.39 \pm 0.29$ |
| GraphPatcher | $63.01 \pm 2.29$ | $57.49 \pm 1.83$ | $45.38 \pm 1.00$ | $67.05 \pm 1.54$ |
| + `Matcha` | $80.99 \pm 0.50$ | $81.38 \pm 0.88$ | $46.78 \pm 1.71$ | $74.46 \pm 0.50$ |

The result are shown in Table 10 above. Although different GNN architectures result in different performance on the target graph, `Matcha` can consistently improve the accuracy. It shows that `Matcha` is compatible with a wide range of GNN architectures.

# D   REPRODUCIBILITY

In this section, we provide details on the datasets, model architecture, and experiment pipelines.

## D.1   DATASETS

We provide more details on the datasets used in the paper, including CSBM synthetic dataset and real-world datasets (Syn-Cora (Zhu et al., 2020), Syn-Products (Zhu et al., 2020), Twitch-E (Rozemberczki et al., 2021), and OGB-Arxiv (Hu et al., 2020)).

- CSBM (Deshpande et al., 2018). We use $N = 5,000$ nodes on both source and target graph with $D = 2,000$ features. Let $\boldsymbol{\mu}_+ = \frac{0.03}{\sqrt{D}} \cdot \mathbf{1}_D$, $\boldsymbol{\mu}_- = -\frac{0.03}{\sqrt{D}} \cdot \mathbf{1}_D$, and $\Delta\boldsymbol{\mu} = \frac{0.02}{\sqrt{D}} \cdot \mathbf{1}_D$.
  - For homo $\leftrightarrow$ hetero, we conduct TTA between CSBM($\boldsymbol{\mu}_+, \boldsymbol{\mu}_-, d = 5, h = 0.8$) and CSBM($\boldsymbol{\mu}_+, \boldsymbol{\mu}_-, d = 5, h = 0.2$).
  - For low $\leftrightarrow$ high, we conduct TTA between CSBM($\boldsymbol{\mu}_+, \boldsymbol{\mu}_-, d = 2, h = 0.8$) and CSBM($\boldsymbol{\mu}_+, \boldsymbol{\mu}_-, d = 10, h = 0.8$).
  - When there are additional attribute shift, we use $\boldsymbol{\mu}_+, \boldsymbol{\mu}_-$ on the source graph, and replace them with $\boldsymbol{\mu}_+ + \Delta\boldsymbol{\mu}, \boldsymbol{\mu}_- + \Delta\boldsymbol{\mu}$ on the target graph.
- Syn-Cora (Zhu et al., 2020) and Syn-Products (Zhu et al., 2020) are widely used datasets to evaluate model's capability in handling homophly and heterophily. The Syn-Cora dataset is generated with various heterophily ratios based on modified preferential attachment process. Starting from an empty initial graph, new nodes are sequentially added into the graph to ensure the desired heterophily ratio. Node features are further generated by sampling node features from the corresponding class in the real-world Cora dataset. Syn-Products is generated in a similar way. For both dataset, we use $h = 0.8$ as the source graph and $h = 0.2$ as the target graph. We use non-overlapping train-test split over nodes on Syn-Cora to avoid label leakage.
- Twitch-E (Rozemberczki et al., 2021) is a set of social networks, where nodes are Twitch users, and edges indicate friendships. Node attributes are the games liked, location and streaming habits of the user. We use 'DE' as the source graph and 'ENGB' as the target graph. We randomly drop a subset of homophily edges on the target graph to inject degree shift and homophily shift.
- OGB-Arxiv (Hu et al., 2020) is a paper citation network of ARXIV papers, where nodes are ARXIV papers and edges are citations between these papers. Node attributes indicate the subject of each paper. We use a subgraph consisting of papers from 1950 to 2011 as the source graph, 2011 to 2014 as the validation graph, and 2014 to 2020 as the target graph. Similarly, we randomly drop a subset of homophily edges on the target graph to inject degree shift and homophily shift.

Table 11: Statistics of datasets used in our experments

| Dataset | Partition | #Nodes | #Edges | #Features | #Classes | Avg. degree $d$ | Node homophily $h$ |
|---|---|---|---|---|---|---|---|
| Syn-Cora | source | 1,490 | 2,968 | 1,433 | 5 | 3.98 | 0.8 |
|  | validation |  |  |  |  |  | 0.4 |
|  | target |  |  |  |  |  | 0.2 |
| Syn-Products | source | 10,000 | 59,648 | 100 | 10 | 11.93 | 0.8 |
|  | validation |  |  |  |  |  | 0.4 |
|  | target |  |  |  |  |  | 0.2 |
| Twitch-E | source | 9,498 | 76,569 | 3,170 | 2 | 16.12 | 0.529 |
|  | validation | 4,648 | 15,588 |  |  | 6.71 | 0.183 |
|  | target | 7,126 | 9,802 |  |  | 2.75 | 0.139 |
| OGB-Arxiv | source | 17,401 | 15,830 | 128 | 40 | 1.82 | 0.383 |
|  | validation | 41,125 | 18,436 |  |  | 0.90 | 0.088 |
|  | target | 169,343 | 251,410 |  |  | 2.97 | 0.130 |

## D.2 MODEL ARCHITECTURE

- For CSBM, Syn-Cora, Syn-Products, we use GPRGNN with $K = 9$. The featurizer is a linear layer, followed by a batchnorm layer, and then the GPR module. The classifier is a linear layer. The dimension for representation is 32.
- For Twitch-E and OGB-Arxiv, we use GPRGNN with $K = 5$. The dimension for representation is 8 and 128, respectively.
- More architectures. For APPNP, we use similar structure as the GPRGNN, while we adapt the $\alpha$ for the personalized pagerank module. For JKNet, we use 2 layers with 32-dimension hidden representation. We adapt the combination layer. For GCNII, we use 4 layers with 32-dimension hidden representation, and adapt the $\alpha_\ell$ for each layer.

## D.3 COMPUTE RESOURCES

We use single Nvidia Tesla V100 with 32GB memory. However, for the majority of our experiments, the memory usage should not exceed 8GB. We switch to Intel(R) Xeon(R) Gold 6240R CPU @ 2.40GHz when recording the computation time.

## E    MORE DISCUSSION

### E.1    ADDITIONAL RELATED WORKS

Graph neural networks (GNNs) have shown great success in various graph applications (Kipf & Welling, 2017; Velickovic et al., 2017; Rozemberczki et al., 2021; Qiu et al., 2022; 2023; Hu et al., 2020; Wang et al., 2023a; Pareja et al., 2020). In Section 2, we briefly introduced selected related works in test-time adaptation and graph domain adaptation. Here with discuss more related works in test-time adaptation, graph out-of-distribution generalization, and homophily-adaptive GNN models.

**More test-time adaptation.**    Another important category of TTA algorithms, beyond those discussed in the main text, leverages data augmentation (Ding et al., 2022; Fu et al., 2023). These methods apply multiple augmentations to the input and enhance prediction robustness through techniques such as ensembling (Hendrycks et al., 2020) or minimizing marginal entropy (Zhang et al., 2022). In the context of graphs, augmentation can also be utilized for test-time adaptation (Xu et al., 2022; Liu et al., 2024b; Zeng et al., 2024). For instance, GTrans (Jin et al., 2023) and DropEdge (Rong et al., 2020) employ augmentation strategies for this purpose. GTrans adopts DropEdge as an augmentation technique and incorporates a contrastive loss that maximizes the similarity between original nodes and their augmented views while penalizing their similarity to negative samples. However, augmentation-based approaches inevitably introduce significant additional computational costs.

**Graph out-of-distribution generalization** (graph OOD) aims to train a GNN model on the source graph that performs well on the target graph with unknown distribution shifts (Li et al., 2022a). Existing graph OOD methods improve the model generalization by manipulating the source graph (Park et al., 2021; Wu et al., 2022), designing disentangled (Ma et al., 2019; Yang et al., 2020; Yan et al., 2024c; Zeng et al., 2023) or casuality-based (Li et al., 2023; Fan et al., 2024) models, and exploiting various learning strategies (Li et al., 2022b; Zhu et al., 2021b). However, graph OOD methods focus on learning a universal model on source and target graphs, while not addressing model adaption to a specific target graph.

**Homophily and heterophily.**    Most GNN models follow the homophily assumption that neighboring nodes tend to share similar labels (Kipf & Welling, 2017; Velickovic et al., 2017; Yan et al., 2024a). Various message-passing (Wang & Zhang, 2022; Zhu et al., 2021a; Xu et al., 2024; Yan et al., 2024b) and aggregation (Chien et al., 2021; Bo et al., 2021; Zhu et al., 2020; Xu et al., 2018; 2023) paradigms have been proposed to extend GNN models to heterophilic graphs. These GNN structures often embrace additional parameters, e.g., the aggregation weights for GPRGNN (Chien et al., 2021) and H2GCN (Zhu et al., 2020), to handle both homophilic and heterophilic graphs. Such parameters provide the flexibility we need to adapt models to shifted graphs. However, these methods focus on the model design to handle either homophilic or heterophilic graph, without considering distribution shifts.

### E.2    LIMITATIONS

**Assumption on source model.**    Since we mainly focus on the challenge of distribution shifts. Our proposed algorithm assumes that the source model should be able to learn class-clustered representations on the source graph, and should generalize well when there are no distribution shifts. In applications with extremely low signal-to-noise ratio, our algorithm's improvement in accuracy might not be guaranteed. However, we would like to point out that this is a challenge faced by almost all TTA algorithms (Zhao et al., 2023).

**Computational efficiency and scalability.**    Our proposed algorithm introduce additional computational overhead during testing. However, we quantify the additional computation time: it is minimal compared to the GNN inference time. Also, `Matcha` is much more efficient that other graph TTA methods.

### E.3    BROADER IMPACTS

Our paper is foundational research related to test-time adaptation on graph data. It focus on node classification as an existing task. We believe that there are no additional societal consequence that must be specifically highlighted here.

