# OpenReview forum: "Matcha: Mitigating Graph Structure Shifts with Test-Time Adaptation"
_ICLR.cc/2025/Conference — ICLR 2025 Poster_

### Official Review · Reviewer_M4v2 · 2024-10-29

**Soundness:** 3
**Presentation:** 2
**Contribution:** 2
**Rating:** 6
**Confidence:** 1

**Summary:**

[Update]
Thanks for the authors'  feedback. My concerns have been addressed. I raised my vote accordingly.

The authors present a method called AdaRC, which is designed to improve the accuracy of Graph Neural Networks (GNNs) under testing time graph structure shifting conditions. It achieves this by adapting GNN architectures in real-time based on the changing graph structure during testing.  The key idea is to introduce a prediction-informed clustering loss which leads to a better learned node representation. The authors demonstrate the superiority of AdaRC algorithm on several popular benchmark datasets with comparisons to popular baseline methods.

**Strengths:**

* Propose a novel loss (PIC loss) to improve the robustness of learned graph embedding when structure shifting is presented.
* The algorithm comes with theoretical guarantees, although I did not check the correctness of the proof.
* Experiments on both controlled synthetic datasets and real-world datasets show the effectiveness of the proposed algorithm.

**Weaknesses:**

* The paper is not easy to follow, especially for readers without years of experiences in graph neural networks. It would be nice if the authors could provide high level intuitions behind their algorithms, before giving mathematical statements.

* To my best guess, the proposed method is not data driven. The algorithm relies on several theoretical assumptions in order to extract an invariant node embedding on the shifted graph structure. The major concern is that, such assumptions may not hold in real-world tasks. It is also hard to verify whether the assumptions hold or not for a given task. This greatly limits the practical value of the algorithm.

**Questions:**

* If the graph structure is shifting in real-time, that is, there is no stable distribution in testing time, will the proposed algorithm still work well?

---

> ### Author Response · Authors · 2024-11-21
> **Response to Reviewer M4v2 (Part 1)**
>
> Dear Reviewer M4v2,
>
> Thank you for recognizing the novelty of the PIC loss, the theoretical guarantees of AdaRC, and the effectiveness of the algorithm demonstrated through experiments on both synthetic and real-world datasets. Your acknowledgment of these contributions is greatly appreciated. Below, we address your questions and concerns:
>
> # W1. High-level Intuitions of Math
>
> > The paper is not easy to follow, especially for readers without years of experiences in graph neural networks. It would be nice if the authors could provide high level intuitions behind their algorithms, before giving mathematical statements.
>
> Thank you for your suggestion. We agree that providing high-level intuitions alongside mathematical statements can make the paper more accessible. Below, we outline the intuitive explanations behind the key components of our theoretical analysis:
> - **Attribute shifts and structure shifts have different impact patterns (Proposition 3.3 and 3.4)**. We visually illustrated this distinction in Figure 2. Attribute shifts cause a translational change in the overall distribution, where nodes from different classes remain separable. In contrast, structure shifts mix the distributions of node representations from different classes, leading to reduced discriminative power.
> - **Adjusting the hop-aggregation parameter can restore the quality of degraded node representations (Proposition 3.5)**. We provided two examples (lines 267 - 286) to demonstrate how adjusting the hop-aggregation parameter $\gamma$ can mitigate the degradation in node representation quality caused by structure shifts. To further enhance clarity, we included visualizations for these examples in Appendix A.2, and also [this anonymous repo](https://anonymous.4open.science/r/AdaRC-ICLR-Rebuttal-93E4/AdaRC_adapt_gamma.pdf) for your convenience. In the visualizations, the first row represents homophily shifts, while the second row represents degree shifts. The first column shows the source graph, the second column shows the target graph before adjusting $\gamma$, and the third column shows the target graph after adjusting $\gamma$. These visualizations show that while homophily shifts and degree shifts degrade representation quality, tuning $\gamma$ effectively alleviates this degradation, improving the separability of node representations.
>
> # W2. Assumptions
>
> > To my best guess, the proposed method is not data driven. The algorithm relies on several theoretical assumptions in order to extract an invariant node embedding on the shifted graph structure. The major concern is that, such assumptions may not hold in real-world tasks. It is also hard to verify whether the assumptions hold or not for a given task. This greatly limits the practical value of the algorithm.
>
> Thank you for your thoughtful feedback. We would like to address your concern from three perspectives:
> - **CSBM Framework as a Common Analytical Framework**. The CSBM framework is widely used [1,2,3] and captures two key types of shifts we consider: homophily shift and degree shift. It provides an interpretable way to analyze these important challenges.
> - **AdaRC does not depend on CSBM Assumptions**. Our algorithm does not rely on CSBM. We use it solely for theoretical analysis to explain why attribute and structure shifts impact GNNs differently and why adjusting $\gamma$ can restore representation qualities. This does not restrict AdaRC to CSBM scenarios. In practice, AdaRC only requires the graph to exhibit structure shifts, which are common and can often be identified through task knowledge or a validation set.
> - **Verification on Real-World Datasets**. We have also tested our algorithm on various real-world datasets in Table 2 and observed that AdaRC consistently demonstrates its effectiveness, supporting the general applicability of our approach.
>
> [1] Yao Ma, Xiaorui Liu, Neil Shah, Jiliang Tang: Is Homophily a Necessity for Graph Neural Networks? ICLR 2022
>
> [2] Haitao Mao, Zhikai Chen, Wei Jin, Haoyu Han, Yao Ma, Tong Zhao, Neil Shah, Jiliang Tang: Demystifying Structural Disparity in Graph Neural Networks: Can One Size Fit All? NeurIPS 2023
>
> [3] Yujun Yan, Milad Hashemi, Kevin Swersky, Yaoqing Yang, Danai Koutra: Two Sides of the Same Coin: Heterophily and Oversmoothing in Graph Convolutional Neural Networks. ICDM 2022

---

> > ### Author Response · Authors · 2024-11-21
> > **Response to Reviewer M4v2 (Part 2)**
> >
> > # Q1. Real-Time Shift
> >
> > > If the graph structure is shifting in real-time, that is, there is no stable distribution in testing time, will the proposed algorithm still work well?
> >
> > Thank you for raising this question. While our experiments in the paper mainly focused on adapting to a single graph with a stable distribution, we also evaluated AdaRC’s performance on a stream of graphs with evolving structure shifts. Specifically, we used the Syn-Cora dataset, where the model was pre-trained on a source graph with a homophily of 0.8. It was then sequentially adapted to five target graphs with homophilies of 0.1, 0.7, 0.3, 0.9, and 0.2, simulating continuously changing homophily. The experimental setup, except for the sequence of target graphs, was identical to that in the Syn-Cora experiments described in the main text.
> >
> > For example, in the column corresponding to the target graph with a homophily of 0.3, we compared two scenarios: (1) static: directly adapting from 0.8 → 0.3, and (2) evolving: sequentially adapting through 0.8 → 0.1 → 0.7 → 0.3.
> >
> > Table A. Accuracy of AdaRC on static target graph and evolving target graph (mean ± s.d.)
> > | Method | Setting  | Target graph (h = 0.1)       | Target graph (h = 0.7)       | Target graph (h = 0.3)       | Target graph (h = 0.9)       | Target graph (h = 0.2)       |
> > |--------|----------|-----------------------------|-----------------------------|-----------------------------|-----------------------------|-----------------------------|
> > | ERM    | Static   | 63.19 ± 1.28               | 88.88 ± 0.61               | 71.46 ± 0.62               | 97.15 ± 0.32               | 65.67 ± 0.35               |
> > | AdaRC  | Static   | 79.75 ± 1.04               | 90.57 ± 0.47               | 79.68 ± 0.73               | 97.40 ± 0.28               | 78.96 ± 1.08               |
> > | AdaRC  | Evolving | 79.75 ± 1.04               | 90.65 ± 0.33               | 77.43 ± 0.62               | 97.31 ± 0.42               | 78.26 ± 1.02               |
> >
> > The results, shown in Table A above, indicate that **AdaRC achieves performance on evolving graphs highly comparable to that on static graphs**. This demonstrates AdaRC’s ability to handle dynamic scenarios effectively, even under continuously changing graph structures.

---

### Official Review · Reviewer_KHid · 2024-11-02

**Soundness:** 3
**Presentation:** 3
**Contribution:** 3
**Rating:** 6
**Confidence:** 4

**Summary:**

This paper proposes a method to mitigate the impact of structure shifts on graph neural networks (GNNs) during test-time adaptation (TTA). The core innovation lies in the introduction of the prediction-informed clustering (PIC) loss to enhance node representation quality during TTA, alongside adjusting hop-aggregation parameters to handle structure shifts. The approach is designed to be compatible with existing TTA algorithms and shows promising results.

**Strengths:**

1. The introduction of the PIC loss to improve node representation quality without labels is a promising approach. This loss function helps better separate classes by minimizing intra-class variance and maximizing inter-class variance, providing a reliable alternative to traditional entropy-based methods.

2. AdaRC is a plug&play framework that can be seamlessly integrated into existing TTA algorithms, making the approach highly extensible.

**Weaknesses:**

1. The authors mention that traditional TTA methods rely on the quality of node representations, but there are also existing works in graph TTA and DA that similarly address the impact of structure shifts. It is recommended to further discuss the differences and advantages of AdaRC in comparison to these methods, particularly in handling heterophilic graphs or complex structure shifts.

2. In Section 3.2, the authors discuss the distinct impacts of attribute and structure shifts; however, using a single-layer GCN as an example may not be sufficient. Given that adjusting hop-aggregation parameters typically involves integrating multi-layer GCNs and broader neighborhood information, it would be beneficial to explore the impact of multi-layer GCNs in the model.

3. Given that the process of using the PIC loss to update hop-aggregation parameters intuitively focuses on optimizing the distinction of node representations (i.e., separating node attribute boundaries), it is recommended to further explore the specific mechanisms by which various structural shifts affect GNN performance. This would help to enhance the theoretical depth and rigor of the analysis.

**Questions:**

see weakness

---

> ### Author Response · Authors · 2024-11-21
> **Response to Reviewer KHid (Part 1)**
>
> Dear Reviewer KHiD,
>
> We sincerely thank you for your thoughtful review and for recognizing the strengths of our work, including the introduction of the PIC loss for improving node representation quality, and the extensibility of AdaRC as a plug-and-play framework. Below, we address your concerns and suggestions:
>
> # W1. Comparison with Existing Graph TTA and DA Methods
>
> > The authors mention that traditional TTA methods rely on the quality of node representations, but there are also existing works in graph TTA and DA that similarly address the impact of structure shifts. It is recommended to further discuss the differences and advantages of AdaRC in comparison to these methods, particularly in handling heterophilic graphs or complex structure shifts.
>
> Thank you for raising this important point. We have provided a general discussion of graph domain adaptation (GDA) and graph test-time adaptation (GTTA) methods in the related work section. We are happy to elaborate further and clarify how AdaRC differs from and improves upon these methods:
>
> **Graph domain adaptation** (GDA) aims to transfer knowledge from a labeled source graph to an unlabeled target graph with access to both graphs. Most of the GDA algorithms focus on learning invariant representations over the source and target graphs by adversarial learning or minimizing the distance between source and target. For example, SRGNN [1] minimizes the central moment discrepancy of node representations, and DANE [2] aligns the distribution via adversarial learning. However, these methods do not explicitly differentiate between attribute shifts and structure shifts, treating them as a single issue to be solved through invariant representation learning. Recently, StruRW [3] addresses structure shifts explicitly by reweighting edge weights in the source graph to align with the target graph. While these approaches offer useful insights, they rely heavily on access to both source and target graphs simultaneously, making them inapplicable to TTA scenarios where only the target graph is available during adaptation.
>
> **Graph test-time adaptation** (GTTA) aims to adapt a pre-trained GNN to an unlabeled target graph, without re-accessing the source domain during adaptation. Recent works such as GTrans, SOGA, and GraphPatcher take varied approaches to this problem. GTrans [4] modifies the target graph’s features and adjacency matrix during testing by minimizing a contrastive loss. While flexible and data-centric, it suffers from high computational complexity and a large parameter space. SOGA [5] maximizes mutual information between inputs and outputs and enforces consistency between neighboring or structurally similar nodes. However, SOGA is designed for homophilic graphs and thus performs relatively poorly on heterophilic graphs. GraphPatcher [6] focuses primarily on degree shifts, generating virtual nodes to improve predictions on low-degree nodes. In comparison, AdaRC is designed to handle a broad range of structure shifts, including homophily and degree shifts. Additionally, AdaRC achieves this with better computational efficiency, making it a practical and effective solution for TTA scenarios.
>
> [1] Qi Zhu, Natalia Ponomareva, Jiawei Han, Bryan Perozzi: Shift-Robust GNNs: Overcoming the Limitations of Localized Graph Training data. NeurIPS 2021
>
> [2] Yizhou Zhang, Guojie Song, Lun Du, Shuwen Yang, Yilun Jin: DANE: Domain Adaptive Network Embedding. IJCAI 2019
>
> [3] Shikun Liu, Tianchun Li, Yongbin Feng, Nhan Tran, Han Zhao, Qiang Qiu, Pan Li: Structural Re-weighting Improves Graph Domain Adaptation. ICML 2023
>
> [4] Wei Jin, Tong Zhao, Jiayuan Ding, Yozen Liu, Jiliang Tang, Neil Shah: Empowering Graph Representation Learning with Test-Time Graph Transformation. ICLR 2023
>
> [5] Haitao Mao, Lun Du, Yujia Zheng, Qiang Fu, Zelin Li, Xu Chen, Shi Han, Dongmei Zhang: Source Free Graph Unsupervised Domain Adaptation. WSDM 2024
>
> [6] Mingxuan Ju, Tong Zhao, Wenhao Yu, Neil Shah, Yanfang Ye: GraphPatcher: Mitigating Degree Bias for Graph Neural Networks via Test-time Augmentation. NeurIPS 2023

---

> > ### Author Response · Authors · 2024-11-21
> > **Response to Reviewer KHid (Part 2)**
> >
> > # W2. Analysis of Multi-Layer GCNs
> >
> > > In Section 3.2, the authors discuss the distinct impacts of attribute and structure shifts; however, using a single-layer GCN as an example may not be sufficient. Given that adjusting hop-aggregation parameters typically involves integrating multi-layer GCNs and broader neighborhood information, it would be beneficial to explore the impact of multi-layer GCNs in the model.
> >
> > Thank you for this great suggestion. We agree that exploring the impact of multi-layer GCNs in the model would be beneficial, as their performance can be influenced by more complex structure shifts beyond degree shifts and homophily shifts (e.g., clustering coefficients). As suggested, we have extended our analysis to $K$-hop GCNs, which aggregate information from nodes' own features as well as their 1-hop to $K$-hop neighbors. Detailed theoretical analysis is provided in Appendix A.7, and also in [this anonymous repo](https://anonymous.4open.science/r/AdaRC-ICLR-Rebuttal-93E4/AdaRC_multihop_gcn.pdf) for your convenience. All our propositions naturally generalize to $K$-hop GCNs, effectively capturing these more intricate structure shifts.
> >
> > # W3. Mechanisms of How Structure Shifts Affect GNN Performance
> >
> > > Given that the process of using the PIC loss to update hop-aggregation parameters intuitively focuses on optimizing the distinction of node representations (i.e., separating node attribute boundaries), it is recommended to further explore the specific mechanisms by which various structural shifts affect GNN performance. This would help to enhance the theoretical depth and rigor of the analysis.
> >
> > Thank you for your insightful suggestion. We agree that exploring the mechanisms by which structure shifts affect GNN performance is beneficial. In our work, Proposition 3.4 provides detailed analysis of how structure shifts influence performance. Specifically, the effects of homophily and degree shifts can be isolated by setting $\Delta h$ or $\Delta d$ to 0, allowing for a focused examination of each factor. Furthermore, Proposition 3.5 illustrates how the optimal hop-aggregation parameter $\gamma$ depends on homophily and degree, offering additional theoretical insights into their role in adapting to structural shifts.

---

> ### Author Response · Authors · 2024-11-27
> **Thanks for your response**
>
> Dear Reviewer KHid,
>
> Thank you for your insightful questions and for taking the time to review our work and rebuttal in detail. Below, we provide detailed responses to your questions.
>
> # Q1. Homophilic and Heterophilic Graphs
>
> > As mentioned in the rebuttal, SOGA is designed for homophilic graphs and thus performs relatively poorly on heterophilic graphs. So, how about the proposed SOGA on the homophilic graphs and heterophilic graphs?
>
> We understand your question as asking how our proposed *AdaRC* performs on homophilic and heterophilic graphs.
>
> Our AdaRC algorithm does not rely on any homophily assumption. Instead, it dynamically adapts the hop-aggregation parameter to make node representations more discriminative. This adaptability ensures that AdaRC can handle both homophilic and heterophilic graphs effectively, as long as the chosen GNN backbone has sufficient expressiveness to model these graph types. For instance, when using a single-layer GCN as the backbone, AdaRC learns a positive $\gamma$ for homophilic graphs and a negative $\gamma$ for heterophilic graphs, aligning the aggregation process with the underlying graph structure.
>
> As shown in Table 1 of our experiments, we evaluated AdaRC with both homophilic target graphs (hetero $\to$ homo) and heterophilic target graphs (homo $\to$ hetero). AdaRC demonstrates strong performance in both cases. In contrast, SOGA, being designed specifically for homophilic graphs, shows significantly weaker performance on heterophilic target graphs compared to homophilic ones.
>
>
> # Q2. Mechanisms of How Structure Shifts Affect GNN Performance
>
> > The Mechanisms of How Structure Shifts Affect GNN Performance should be explained.
>
> Thank you for your question. We understand that your inquiry follows up on W3 and seeks a standalone explanation of how *homophily shift* and *degree shift* individually lead to a decline in GNN performance. We are happy to provide a more intuitive explanation.
>
> **Theoretical analysis.** Using the theoretical analysis of a single-layer GCN on a CSBM graph from our paper as an example, for a positive-class sample, its representation distribution can be expressed as:
> $$
> \boldsymbol{z}\_i \sim \mathcal{N} \left( (1 + \gamma h\_i) \boldsymbol{\mu}\_+ + \gamma (1 - h\_i) \boldsymbol{\mu}\_-, \left( 1 + \frac{\gamma^2}{d\_i} \right) \boldsymbol{I} \right)
> $$
>
> - **Homophily shift:** Changes in $h_i$ (node homophily) affect the mean of $\boldsymbol{z}_i$​, while the variance remains constant. This reduces the inter-class distance as the centroids of the two classes move closer together, but the intra-class variance does not change. Consequently, the representations of different classes overlap more, leading to decreased accuracy.
> - **Degree shift:** Changes in $d_i$ (node degree) alter the variance of $\boldsymbol{z}_i$, without affecting the mean. This increases intra-class variance while keeping inter-class distance unchanged. Similarly, this increased overlap between class representations reduces the ability to distinguish between classes and degrades accuracy.
>
> Both shifts lead to overlapping representations between different classes, which negatively impacts classification performance.
>
> **Visualization.** To further illustrate, we provide two visualization examples in Appendix A.2 (also available in [this anonymous repo](https://anonymous.4open.science/r/AdaRC-ICLR-Rebuttal-93E4/AdaRC_adapt_gamma.pdf) for your convenience):
>
> - Under **homophily shift**, the variance of the two class distributions remains constant, but the centroids of the two classes move closer together.
> - Under **degree shift**, the centroids remain unchanged, but the intra-class variance increases.
>
> Both scenarios clearly demonstrate how structure shifts result in greater overlap between class representations, ultimately reducing accuracy.
>
> We hope this explanation provides a clear understanding of the mechanisms. Please let us know if further clarification or additional details are needed.

---

### Official Review · Reviewer_qcfU · 2024-11-03

**Soundness:** 3
**Presentation:** 3
**Contribution:** 3
**Rating:** 6
**Confidence:** 4

**Summary:**

This paper proposes AdaRC, a novel framework aimed at improving graph neural networks (GNNs) under distribution shifts, particularly those involving changes in graph structure. Traditional test-time adaptation (TTA) methods are typically designed to handle shifts in node attributes, but they perform poorly with structure shifts, where connectivity patterns vary significantly between training and test data. AdaRC addresses this gap by dynamically adjusting the hop-aggregation parameters in GNNs, effectively enhancing node representation quality under structure shifts. Additionally, a prediction-informed clustering (PIC) loss is introduced to encourage clear separations between node categories, further improving adaptation. Extensive experiments demonstrate AdaRC's compatibility with existing TTA methods, yielding notable performance improvements on both synthetic and real-world datasets.

**Strengths:**

- AdaRC introduces a new approach to handling structure shifts by adapting hop-aggregation parameters, which allows the model to improve node representation quality in ways traditional TTA methods cannot.

- The experimental evaluation is comprehensive, covering both synthetic and real-world datasets, and demonstrates substantial improvements in accuracy and robustness across different shifts, particularly structure shifts.

- The theoretical insights into the impact of structure and attribute shifts are clearly explained, and the figures and visualizations effectively illustrate the method's impact.

**Weaknesses:**

- While the PIC loss offers advantages, its application may face scalability challenges with very large graphs due to the computational overhead introduced by clustering operations. The method's efficiency under extreme graph sizes could be explored further.

- AdaRC's performance relies on the optimization of hop-aggregation parameters, which may not generalize across all GNN architectures without tuning. This reliance might limit its applicability for varied and complex GNN models.

- AdaRC has been evaluated on static graphs with controlled shifts. Its applicability to dynamic graphs, where structure shifts occur over time, is not explored, which could be crucial for certain real-time applications.

- The performance of AdaRC's clustering approach may be sensitive to the quality of initial predictions, particularly in cases where pseudo-labeling accuracy is low, potentially affecting the model’s stability in scenarios with noisy initial data.

**Questions:**

- Can AdaRC be adapted to handle continuously evolving graphs where node connectivity changes over time?

- How does the computational efficiency of AdaRC scale when applied to graphs with millions of nodes?

- Could PIC loss performance be enhanced by integrating more robust initialization methods for pseudo-labeling in low-confidence scenarios?

- Does the framework support automated tuning of hop-aggregation parameters across different GNN architectures, or would manual tuning be required for optimal results?

---

> ### Author Response · Authors · 2024-11-21
> **Response to Reviewer qcfU (Part 1)**
>
> Dear Reviewer qcfU,
>
> We sincerely thank you for your thoughtful review and for recognizing the strengths of our work, including AdaRC's novel approach to addressing structure shifts, its compatibility with existing TTA methods, the clarity of our theoretical insights, and the comprehensiveness of our experimental evaluations. Below, we address the concerns and questions raised in your review:
>
> # W1 & Q2. Scalability
>
> > W1. While the PIC loss offers advantages, its application may face scalability challenges with very large graphs due to the computational overhead introduced by clustering operations. The method's efficiency under extreme graph sizes could be explored further.
>
> > Q2. How does the computational efficiency of AdaRC scale when applied to graphs with millions of nodes?
>
> Thank you for raising this important point. We would like to clarify that AdaRC does not perform additional clustering operations. Instead, it leverages the pseudo-labels from the BaseTTA method as the clustering results. This design ensures that the computational overhead of calculating the PIC loss is linear with respect to the number of nodes in the graph.
>
> To further validate this, we have conducted experiments on graphs of varying sizes from 1 million to 10 million nodes. The result in given in Figure 12 of Appendix C.8, and also in [this anonymous repo](https://anonymous.4open.science/r/AdaRC-ICLR-Rebuttal-93E4/AdaRC_scalability.pdf) for your convenience. The results confirm that the computational time for AdaRC indeed scales linearly with graph size, demonstrating its efficiency even for large graphs.
>
> # W2 & Q4. Automated Tuning of Hop-Aggregation Parameters
>
> > W2. AdaRC's performance relies on the optimization of hop-aggregation parameters, which may not generalize across all GNN architectures without tuning. This reliance might limit its applicability for varied and complex GNN models.
>
> > Q4. Does the framework support automated tuning of hop-aggregation parameters across different GNN architectures, or would manual tuning be required for optimal results?
>
> For the GNNs discussed in the paper (GPRGNN, APPNP, GCNII, JKNet), AdaRC supports automated tuning of hop-aggregation parameters by minimizing the PIC loss, without requiring any manual tuning. Specifically, while the original implementations of these frameworks may treat hop-aggregation parameters as hyperparameters (e.g., the teleport probability $\alpha$ in APPNP), we simply treat these parameters as trainable parameters. This allows us to optimize them automatically through gradient descent to minimize the PIC loss, effectively enabling automated adjustment of hop-aggregation parameters.
>
> # W3 & Q1. Performance on Evolving Graphs
>
> > W3. AdaRC has been evaluated on static graphs with controlled shifts. Its applicability to dynamic graphs, where structure shifts occur over time, is not explored, which could be crucial for certain real-time applications.
>
> > Q1. Can AdaRC be adapted to handle continuously evolving graphs where node connectivity changes over time?
>
> While our experiments in the paper mainly focused on adapting to a single graph with a stable distribution, we also evaluated AdaRC’s performance on a stream of graphs with evolving structure shifts. Specifically, we used the Syn-Cora dataset, where the model was pre-trained on a source graph with a homophily of 0.8. It was then sequentially adapted to five target graphs with homophilies of 0.1, 0.7, 0.3, 0.9, and 0.2, simulating continuously changing homophily. The experimental setup, except for the sequence of target graphs, was identical to that in the Syn-Cora experiments described in the main text.
>
> For example, in the column corresponding to the target graph with a homophily of 0.3, we compared two scenarios: (1) static: directly adapting from 0.8 → 0.3, and (2) evolving: sequentially adapting through 0.8 → 0.1 → 0.7 → 0.3.
>
> Table A. Accuracy of AdaRC on static target graph and evolving target graph (mean ± s.d.)
> | Method | Setting  | Target graph (h = 0.1)       | Target graph (h = 0.7)       | Target graph (h = 0.3)       | Target graph (h = 0.9)       | Target graph (h = 0.2)       |
> |--------|----------|-----------------------------|-----------------------------|-----------------------------|-----------------------------|-----------------------------|
> | ERM    | Static   | 63.19 ± 1.28  | 88.88 ± 0.61  | 71.46 ± 0.62 | 97.15 ± 0.32               | 65.67 ± 0.35               |
> | AdaRC  | Static   | 79.75 ± 1.04 | 90.57 ± 0.47  | 79.68 ± 0.73 | 97.40 ± 0.28               | 78.96 ± 1.08               |
> | AdaRC  | Evolving | 79.75 ± 1.04 | 90.65 ± 0.33 | 77.43 ± 0.62 | 97.31 ± 0.42               | 78.26 ± 1.02               |
>
> The results, shown in Table A above, indicate that **AdaRC achieves performance on evolving graphs highly comparable to that on static graphs**. This demonstrates AdaRC’s ability to handle dynamic scenarios effectively, even under continuously changing graph structures.

---

> > ### Author Response · Authors · 2024-11-21
> > **Response to Reviewer qcfU (Part 2)**
> >
> > # W4 & Q3. Initial Prediction
> >
> > > W4. The performance of AdaRC's clustering approach may be sensitive to the quality of initial predictions, particularly in cases where pseudo-labeling accuracy is low, potentially affecting the model’s stability in scenarios with noisy initial data.
> >
> > Thank you for raising this concern. We would like to highlight that we addressed this issue in Appendix C.2, where we provided an example demonstrating AdaRC’s robustness to noisy initial predictions. Specifically:
> > - Before adaptation, the predictions exhibit significant noise, with substantial overlap in the logits of the two classes, reflecting low pseudo-labeling accuracy.
> > - During adaptation, despite the noisy initialization, AdaRC is able to gradually refine the node representations by leveraging the PIC loss, leading to improved class separation and accuracy.
> >
> > This example illustrates that AdaRC can effectively handle scenarios with noisy initial data, maintaining stability and progressively enhancing model performance.
> >
> > > Q3. Could PIC loss performance be enhanced by integrating more robust initialization methods for pseudo-labeling in low-confidence scenarios?
> >
> > Thank you for this great point. Indeed, the performance of PIC loss can be further enhanced with more robust pseudo-labeling methods. More accurate pseudo-labels from BaseTTA lead to gradients that better optimize node representations, resulting in improved performance. In our experiments, stronger BaseTTA methods typically also yield better results when combined with AdaRC.

---

### Official Review · Reviewer_eWx1 · 2024-11-03

**Soundness:** 3
**Presentation:** 3
**Contribution:** 3
**Rating:** 6
**Confidence:** 2

**Summary:**

This work proposes the AdaRC framework, a test-time adaptation approach for graph neural networks (GNNs). At the core of AdaRC is a prediction-informed clustering (PIC) loss, designed to seamlessly integrate with existing methods. Experimental results demonstrate the effectiveness of PIC in enhancing GNN performance.

**Strengths:**

The paper analyzes the performance gap in a simple single-layer Graph Convolutional Network (GCN) using graphs generated from the Community Structure Benchmark Model (CSBM). It effectively investigates the impacts of attribute shifts and structural shifts on performance.

The proposed PIC loss is both simple and effective, allowing for seamless integration with existing frameworks and methods. Its effectiveness is validated through experimental results.

**Weaknesses:**

It is worth noting that, as I am not deeply familiar with this specific area, my comments here may lack technical depth. I will rely on the insights from other expert reviewers and will focus primarily on their evaluations. Here, a few potential concerns include:

- The performance improvements on the Twitch-E and OGB-Arxiv datasets appear marginal. Could the authors provide further analysis to clarify the reasons behind these limited gains?

- The gains achieved seem to vary significantly across different methods and backbone models. For instance, the Tent method demonstrates the highest improvements. Could the authors offer insights into why this variation occurs?

**Questions:**

See weakness.

---

> ### Author Response · Authors · 2024-11-21
> **Response to Reviewer eWx1**
>
> Dear Reviewer eWx1,
>
> We sincerely thank you for your thoughtful review and for highlighting the strengths of our work, including the effectiveness of the proposed PIC loss and its seamless integration with existing frameworks, as well as the comprehensive analysis of attribute and structure shifts using CSBM. Below, we address your concerns and questions:
>
> # W1. Performance on Real-World Graphs
>
> > The performance improvements on the Twitch-E and OGB-Arxiv datasets appear marginal. Could the authors provide further analysis to clarify the reasons behind these limited gains?
>
> The relatively lower performance on Twitch-E and OGB-Arxiv datasets is due to the higher uncertainty and more complex distribution shifts present in these datasets, which make the classification task harder. This observation is consistent with results in related works [1]. We also experimented with using cross-entropy loss instead of PIC loss. Even when the model was aware of 1% of the testing labels, the accuracy achieved was only 58.21% on Twitch and 42.12% on Arxiv. In comparison, our PIC loss achieved 56.76% and 41.74% accuracy, respectively, without requiring any testing labels.
>
> [1] Wei Jin, Tong Zhao, Jiayuan Ding, Yozen Liu, Jiliang Tang, Neil Shah: Empowering Graph Representation Learning with Test-Time Graph Transformation. ICLR 2023
>
> # W2. Gains Achieved Vary
>
> > The gains achieved seem to vary significantly across different methods and backbone models. For instance, the Tent method demonstrates the highest improvements. Could the authors offer insights into why this variation occurs?
>
> Thank you for your observation. The variation in gains across different TTA methods primarily arises from the distinct designs of these algorithms, which make them suitable for different types of attribute shifts. For example, Tent adjusts the parameters of the batch normalization layer, making it particularly effective for handling attribute shifts that resemble large translational changes. On the other hand, T3A updates class prototypes using testing data, which may be more effective for complex shift patterns with smaller magnitudes. We believe that selecting the most appropriate TTA algorithm based on the specific nature of the attribute shift is an interesting direction for future research.

---

### Official Review · Reviewer_jAsf · 2024-11-03

**Soundness:** 3
**Presentation:** 3
**Contribution:** 2
**Rating:** 6
**Confidence:** 3

**Summary:**

This paper addresses the issue of structure shifts in Graph Neural Networks at test time.  The authors claim that existing test-time adaptation methods are primarily designed for attribute shifts rather than structure shifts. Based on theoretical analysis, they propose a framework called AdaRC, which aggregates hop parameters to restore good node representation, alleviating the negative impact of structure shifts. They also employ pseudo-class labels to optimize the hop-aggregation parameters.

**Strengths:**

1. The paper points out the problem of structure shifts, which is indeed overlooked in previous research on TTA, or not focused explicitly at least.

2. The proposed AdaRC framework provides a simple yet effective solution for graphs with structure shifts at test time. It is compatible with existing Test-Time Adaptation algorithms, which allows better adaption.

3. The authors conduct a detailed theoretical analysis of the impact of structure shifts on a single-layer GCN on a CSBM-generated graph.

**Weaknesses:**

1. The theoretical analysis focuses on single-layer GCNs and CSBM-generated graphs, which represent a simplified model where structure shifts are easily identified. However, the evolving real-world graph structures are far more complex. It is unclear whether the findings on this simplified model still hold for real-world graphs, particularly in domains where the task accuracy is not high.

2. The experimental results, which show promising performance, are primarily based on synthetic datasets or real-world datasets with synthetic structure shifts. The verification that the structure shifts existence in the real-world graphs should be included.

3. The experimental setup for real-world datasets lacks transparency. The paper mentions injecting structure shifts by deleting a subset of homophilic edges. However, it did not specify the quantity of this deletion and the selection criteria.

4. The paper presents full results for only one GNN backbone. Presenting results for different GNN backbones, especially fundamental ones, would provide a more comprehensive understanding of AdaRC's effectiveness and applicability across various architectures.

**Questions:**

1. AdaRC is a plug-and-play method, have you explored scenarios where it might conflict with other TTA methods? For instance, if a TTA method focuses on adapting to changes in node features while AdaRC adjusts to structure shifts, could their optimization goals lead them in opposite directions?

2. On Twitch-E and OGB-Arxiv, AdaRC shows only minor improvements without artificially injected structure shifts. Does this suggest that the impact of structure shift might be less significant in these real-world datasets?  Could this be because the type of structure shift addressed by AdaRC is less prevalent in these contexts?

---

> ### Author Response · Authors · 2024-11-21
> **Response to Reviewer jAsf (Part 1)**
>
> Dear Reviewer jAsf,
>
> We sincerely thank you for your thoughtful review and for recognizing the strengths of our work, including the identification of structure shifts as an overlooked problem, the simplicity and effectiveness of the AdaRC framework, and the detailed theoretical analysis provided. Below, we address the concerns and questions raised in your review:
>
> # W1. Simplified Model for Theoretical Analysis
>
> > The theoretical analysis focuses on single-layer GCNs and CSBM-generated graphs, which represent a simplified model where structure shifts are easily identified. However, the evolving real-world graph structures are far more complex. It is unclear whether the findings on this simplified model still hold for real-world graphs, particularly in domains where the task accuracy is not high.
>
> We acknowledge that our theoretical analysis primarily focuses on single-layer GCNs and CSBM-generated graphs. This choice was intentional to provide a rigorous and interpretable foundation for understanding the impact of common structure shifts, including degree and homophily shifts.
>
> We agree that real-world graphs exhibit structure shifts beyond homophily and degree, such as clustering coefficients, which can also impact the performance of multi-layer GNNs. To address this, we have further extended our analysis to $K$-hop GCNs, which aggregate information from nodes' own features as well as their 1-hop to $K$-hop neighbors. Detailed theoretical analysis is provided in Appendix A.7, and also [this anonymous repo](https://anonymous.4open.science/r/AdaRC-ICLR-Rebuttal-93E4/AdaRC_multihop_gcn.pdf) for your convenience. All our propositions naturally generalize to $K$-hop GCNs, capturing more complex structure shifts. Additionally, our analysis does not rely on assumptions about task accuracy, making it broadly applicable.
>
> # W2. Verification of Structure Shifts in Real-World Graphs
>
> > The experimental results, which show promising performance, are primarily based on synthetic datasets or real-world datasets with synthetic structure shifts. The verification that the structure shifts existence in the real-world graphs should be included.
>
> Thank you for your suggestion. We agree that verifying the presence of structure shifts in real-world graphs is important and beneficial. Previous studies [1,2] have analyzed common real-world graphs and reported graph metrics such as homophily, degree, demonstrating the prevalence of structure shifts. Additionally, in our work, we have included the metrics for the datasets we used (see Appendix D.1) to provide further insights.
>
> If there is a specific graph dataset that you are particularly interested in, we would be happy to calculate and provide the relevant graph metrics to verify the presence of structure shifts.
>
> [1] Eli Chien, Jianhao Peng, Pan Li, Olgica Milenkovic: Adaptive Universal Generalized PageRank Graph Neural Network. ICLR 2021
>
> [2] Shikun Liu, Tianchun Li, Yongbin Feng, Nhan Tran, Han Zhao, Qiang Qiu, Pan Li: Structural Re-weighting Improves Graph Domain Adaptation. ICML 2023
>
> # W3. Experiment Setup Details
>
> > The experimental setup for real-world datasets lacks transparency. The paper mentions injecting structure shifts by deleting a subset of homophilic edges. However, it did not specify the quantity of this deletion and the selection criteria.
>
> Thank you for your comment. Due to page limits, detailed experimental setup is provided in Appendix D.1, including the number of edges left. Specifically, the selection of homophilic edges for deletion is performed using random sampling without replacement.
>
> # W4. GNN Backbones
>
> > The paper presents full results for only one GNN backbone. Presenting results for different GNN backbones, especially fundamental ones, would provide a more comprehensive understanding of AdaRC's effectiveness and applicability across various architectures.
>
> Thank you for your valuable suggestion. We agree that full experiments with other GNN architectures can provide more comprehensive understandings. Following your advice, we have conducted full experiments with different GNN backbones on Syn-Cora. We include all BaseTTAs used in our paper, and their combination with AdaRC. The results are updated in Appendix C.9. The results show that, although different GNN architectures yield varying levels of performance on the target graph, AdaRC consistently enhances accuracy across all cases. This demonstrates that AdaRC is compatible with a wide range of GNN architectures, highlighting its broad applicability and effectiveness.

---

> > ### Author Response · Authors · 2024-11-21
> > **Response to Reviewer jAsf (Part 2)**
> >
> > # Q1. Conflict between AdaRC and BaseTTA
> >
> > > AdaRC is a plug-and-play method, have you explored scenarios where it might conflict with other TTA methods? For instance, if a TTA method focuses on adapting to changes in node features while AdaRC adjusts to structure shifts, could their optimization goals lead them in opposite directions?
> >
> > Thank you for the insightful question. Usually, AdaRC and other TTA methods are orthogonal in their design. AdaRC focuses on improving the quality of node representations by addressing structure shifts, while TTA methods typically focus on better utilizing the node representations. This distinction minimizes the likelihood of conflicting optimization goals.
> >
> > In our experiments, we did not observe any cases where combining AdaRC with a BaseTTA method, both of which improve performance individually, resulted in a performance drop. However, we agree that using a stronger BaseTTA usually leads to better overall performance.
> >
> > # Q2. Performance on Real-World Graphs
> >
> > > On Twitch-E and OGB-Arxiv, AdaRC shows only minor improvements without artificially injected structure shifts. Does this suggest that the impact of structure shift might be less significant in these real-world datasets? Could this be because the type of structure shift addressed by AdaRC is less prevalent in these contexts?
> >
> > The relatively lower performance on Twitch-E and OGB-Arxiv datasets is due to the higher uncertainty and more complex distribution shifts present in these datasets, which make the classification task harder. This observation is consistent with results in related works [3]. We also experimented with using cross-entropy loss instead of PIC loss. Even when the model was aware of 1% of the testing labels, the accuracy achieved was only 58.21% on Twitch and 42.12% on Arxiv. In comparison, our PIC loss achieved 56.76% and 41.74% accuracy, respectively, without requiring any testing labels.
> >
> > [3] Wei Jin, Tong Zhao, Jiayuan Ding, Yozen Liu, Jiliang Tang, Neil Shah: Empowering Graph Representation Learning with Test-Time Graph Transformation. ICLR 2023

---

### Author Response · Authors · 2024-11-21
**Response to All Reviewers (Part 1)**

Dear Reviewers,

We sincerely thank you for your valuable feedback and for recognizing the strengths of our work. Below, we summarize the main contributions and key strengths of our paper:
- **Addressing an Important Yet Overlooked Problem** (jAsf).  Our work focuses on the critical yet previously overlooked problem of structure shifts in graph data, in the context of TTA.
- **Detailed Theoretical Analysis** (jAsf, eWx1, qcfU). We provide rigorous theoretical analysis of the impact of structure shifts on single-layer GCNs using CSBM graphs, offering deep insights into the challenges posed by these shifts.
- **Simple, Effective, and Compatible Method** (all reviewers). Our proposed AdaRC framework is simple yet effective in improving node representation quality and addressing structure shifts. It is fully compatible with existing TTA algorithms, enabling broader applicability and enhanced performance.
- **Comprehensive Experiments** (qcfU, M4v2). Extensive experiments on both synthetic and real-world datasets validate the robustness and effectiveness of AdaRC across various types of shifts, particularly structure shifts.

We deeply appreciate your constructive suggestions and would like to address some common questions raised about our work

# 1. Gap between CSBM and Real-World Graphs (jAsf, KHid)

Our theoretical analysis primarily focuses on single-layer GCNs and CSBM graphs, which effectively capture homophily and degree shifts, two factors commonly observed in real-world scenarios that significantly influence GNN performance. Furthermore, reviewers' questions have inspired us to further extend our theoretical framework to multi-hop GCNs (in Appendix A.7, and also in [this anonymous repo](https://anonymous.4open.science/r/AdaRC-ICLR-Rebuttal-93E4/AdaRC_multihop_gcn.pdf)), allowing us to analyze more complex structure shifts beyond single-layer settings.
- **CSBM Reflects Real-World Structure Shifts**. CSBM graphs have been frequently utilized in theoretical studies [1,2,3] of GNNs due to their ability to model two critical graph metrics, degree and homophily, that significantly influence GNN performance. We believe CSBM effectively captures the challenges posed by structure shifts commonly encountered in real-world applications.
- **Extending Analysis to Multi-Hop GCNs**. We agree that real-world graphs exhibit structure shifts beyond homophily and degree, such as clustering coefficients, which can also impact the performance of multi-layer GNNs. To address this, we have extended our analysis to $K$-hop GCNs, which aggregate information from nodes' own features as well as their 1-hop to $K$-hop neighbors. Detailed theoretical analysis is provided in Appendix A.7 and also in [this anonymous repo](https://anonymous.4open.science/r/AdaRC-ICLR-Rebuttal-93E4/AdaRC_multihop_gcn.pdf) for your convenience. All our propositions naturally generalize to $K$-hop GCNs, capturing more complex structure shifts.

[1] Yao Ma, Xiaorui Liu, Neil Shah, Jiliang Tang: Is Homophily a Necessity for Graph Neural Networks? ICLR 2022

[2] Haitao Mao, Zhikai Chen, Wei Jin, Haoyu Han, Yao Ma, Tong Zhao, Neil Shah, Jiliang Tang: Demystifying Structural Disparity in Graph Neural Networks: Can One Size Fit All? NeurIPS 2023

[3] Yujun Yan, Milad Hashemi, Kevin Swersky, Yaoqing Yang, Danai Koutra: Two Sides of the Same Coin: Heterophily and Oversmoothing in Graph Convolutional Neural Networks. ICDM 2022

# 2. Performance on Real-World Graphs (jAsf, eWx1)
The relatively lower performance on Twitch-E and OGB-Arxiv datasets is due to the higher uncertainty and more complex distribution shifts present in these datasets, which make the classification task harder. This observation is consistent with results in related works [4]. We also experimented with using cross-entropy loss instead of PIC loss. Even when the model was aware of 1\% of the testing labels, the accuracy achieved was only 58.21\% on Twitch and 42.12\% on Arxiv. In comparison, our PIC loss achieved 56.76\% and 41.74\% accuracy, respectively, without requiring any testing labels.

[4] Wei Jin, Tong Zhao, Jiayuan Ding, Yozen Liu, Jiliang Tang, Neil Shah: Empowering Graph Representation Learning with Test-Time Graph Transformation. ICLR 2023

---

> ### Author Response · Authors · 2024-11-21
> **Response to All Reviewers (Part 2)**
>
> # 3. Performance on Evolving Graphs (qcfU, M4v2)
>
> While our experiments in the paper mainly focused on adapting to a single graph with a stable distribution, we also evaluated AdaRC’s performance on a stream of graphs with evolving structure shifts. Specifically, we used the Syn-Cora dataset, where the model was pre-trained on a source graph with a homophily of 0.8. It was then sequentially adapted to five target graphs with homophilies of 0.1, 0.7, 0.3, 0.9, and 0.2, simulating continuously changing homophily. The experimental setup, except for the sequence of target graphs, was identical to that in the Syn-Cora experiments described in the main text.
>
> For example, in the column corresponding to the target graph with a homophily of 0.3, we compared two scenarios: (1) static: directly adapting from 0.8 → 0.3, and (2) evolving: sequentially adapting through 0.8 → 0.1 → 0.7 → 0.3.
>
> Table A. Accuracy of AdaRC on static target graph and evolving target graph (mean ± s.d.)
> | Method | Setting  | Target graph (h = 0.1)       | Target graph (h = 0.7)       | Target graph (h = 0.3)       | Target graph (h = 0.9)       | Target graph (h = 0.2)       |
> |--------|----------|-----------------------------|-----------------------------|-----------------------------|-----------------------------|-----------------------------|
> | ERM    | Static   | 63.19 ± 1.28               | 88.88 ± 0.61               | 71.46 ± 0.62               | 97.15 ± 0.32               | 65.67 ± 0.35               |
> | AdaRC  | Static   | 79.75 ± 1.04               | 90.57 ± 0.47               | 79.68 ± 0.73               | 97.40 ± 0.28               | 78.96 ± 1.08               |
> | AdaRC  | Evolving | 79.75 ± 1.04               | 90.65 ± 0.33               | 77.43 ± 0.62               | 97.31 ± 0.42               | 78.26 ± 1.02               |
>
> The results, shown in Table A above, indicate that **AdaRC achieves performance on evolving graphs highly comparable to that on static graphs**. This demonstrates AdaRC’s ability to handle dynamic scenarios effectively, even under continuously changing graph structures.

---

### Author Response · Authors · 2024-12-01
**Friendly Reminder Regarding Rebuttal Discussion**

Dear Reviewers,

Thank you for taking the time to provide thoughtful feedback on our submission, "AdaRC: Mitigating Graph Structure Shifts during Test-Time." We’ve greatly benefited from your comments and hope our rebuttal addressed your questions and concerns effectively.

As December 2 is the final day to post feedback to the authors, we wanted to kindly remind you that we are happy to engage in further discussions or clarify any remaining points if needed.

We appreciate your effort in reviewing our work and look forward to hearing your thoughts.

Best regards,

The Authors

---

### Author Response · Authors · 2024-12-03
**Brief Summary of Contributions and Modifications**

We sincerely thank all reviewers for their constructive feedback and suggestions, which have greatly improved the quality of our paper. As the discussion period concludes, we would like to provide a brief summary of our paper’s contributions and the modifications made during the rebuttal:

# Contributions
1. **Addressing an Important Yet Overlooked Problem**: Our work focuses on the critical yet underexplored issue of structure shifts in graph data within the context of TTA.
2. **Detailed Theoretical Analysis**: We present rigorous theoretical analysis, uncovering the distinct impact patterns of attribute and structure shifts on GNNs.
3. **Simple, Effective, and Compatible Method**: Our proposed AdaRC framework is simple yet effective in improving node representation quality under structure shifts. It is fully compatible with existing TTA algorithms, enabling broader applicability and enhanced performance.
4. **Comprehensive Experiments**: Extensive experiments on synthetic and real-world datasets validate the robustness and effectiveness of AdaRC across various types of shifts, particularly structure shifts.

# Rebuttal Modifications
1. **Intuitive Examples**: To better illustrate our theoretical findings, we added visualizations in Appendix A.2.
2. **Extension of Theory**: We extended our theoretical analysis to multi-hop GCNs, to analyze the impact of structure shifts beyond homophily and degree shifts, as detailed in Appendix A.7.
3. **Additional Experiments**:
    1. **Scalability**: We evaluated AdaRC’s computational cost with increasing graph sizes (Appendix C.8).
    2. **More Architectures**: We provided additional full experiments combining AdaRC with various GNN architectures (Appendix C.9).
    3. **Evolving Graphs**: We tested AdaRC on streams of graphs with evolving structure shifts, with results presented in Table A of [this response](https://openreview.net/forum?id=EpgoFFUM2q&noteId=1aOTie6P7S).

We would also like to respectfully clarify that Reviewer M4v2's evaluation may partially stem from a misunderstanding of our theoretical analysis, as their stated confidence level is only 1. Specifically, they suggested that our algorithm relies on theoretical assumptions. However, AdaRC does not depend on these assumptions; rather, we use CSBM as a common analytical framework to study the challenges of homophily shift and degree shift. To demonstrate the generalizability of AdaRC, we have conducted extensive experiments on real-world datasets, as shown in Table 2.

That said, we are grateful for Reviewer M4v2's suggestion regarding high-level intuitions, which prompted us to include additional visualizations in Appendix A.2 to enhance accessibility and understanding.

Once again, we deeply appreciate the time and effort each reviewer has invested in providing valuable feedback.

---

### Public Comment · ~Wenxuan_Bao1 · 2025-02-12
**Algorithm Name Change Notification**

We have changed the algorithm name from **AdaRC** to **Matcha**. Please note that all previous reviews and rebuttals refer to the old name. We hope this clarification prevents any confusion.

---

### Meta-Review · Area_Chair_4mqY · 2024-12-17

**Metareview:**

This paper proposes AdaRc for solving the structure shifts in graphs, which can adjust hop-aggregation parameters in GNN during test-time adaptation.

The main strengths of the paper include: 1) study an overlooked problem of TTA; 2) the proposed AdaRc is simple yet effective; and 3) detailed theoretical analysis is provided.

Despite these, the reviewers also raised several drawbacks and concerns for this paper in the first round, including: 1) practicality in real-world; 2) experiments on real-world graphs, more GNN backbones and multi-layers; and 3) explanations for the gains among different datasets.

The authors have provided a rebuttal. Only one reviewer has attended the discussion-phase and all the reviewers kept their original ratings after rebuttal, i.e., four weak accepts. The AC has carefully checked the comments of the reviewers and the response of the authors. The AC thinks most of the concerns of the reviewers have been solved. Thus, considering the strengths of this paper, the AC thinks this paper could be a valuable work for TTA with graphs and thus recommends acceptance to this paper.

**Additional Comments On Reviewer Discussion:**

The authors have provided a rebuttal. The reviewers did not change their original positive ratings.

---

### Decision · Program_Chairs · 2025-01-22

Accept (Poster)